# Graph Neural Networks Can (Often) Count Substructures

**Paolo Pellizzoni, Till Hendrik Schulz, Karsten Borgwardt**
Max Planck Institute of Biochemistry, Martinsried, Germany
`{pellizzoni, tschulz, borgwardt}@biochem.mpg.de`

## Abstract

Message passing graph neural networks (GNNs) are known to have limited expressive power in their ability to distinguish some non-isomorphic graphs. Because of this, it is well known that they are unable to detect or count arbitrary graph substructures (i.e., solving the subgraph isomorphism problem), a task that is of great importance for several types of graph-structured data. However, we observe that GNNs are in fact able to count graph patterns quite accurately across several real-world graph datasets. Motivated by this observation, we provide an analysis of the subgraph-counting capabilities of GNNs beyond the worst case, deriving several sufficient conditions for GNNs to be able to count subgraphs and, more importantly, to be able to *sample-efficiently learn* to count subgraphs. Moreover, we develop novel dynamic programming algorithms for solving the subgraph isomorphism problem on restricted classes of pattern and target graphs, and show that message-passing GNNs can efficiently simulate these dynamic programs. Finally, we empirically validate that our sufficient conditions for GNNs to count subgraphs hold on many real-world datasets, providing a theoretically-grounded explanation to our motivating observations.

## 1 Introduction

Graph neural networks (GNNs) have emerged as powerful tools for learning on graph-structured data, achieving significant empirical success across diverse domains including computational chemistry, bioinformatics, and social network analysis. However, the expressivity of these models in the context of graph classification, i.e., their ability to distinguish non-isomorphic graphs, is intrinsically limited by the capabilities of the Weisfeiler-Leman (1-WL) algorithm (Weisfeiler & Leman, 1968), a heuristic used for the graph isomorphism problem (Morris et al., 2019; Xu et al., 2018). 1-WL is known to fail to distinguish certain classes of graphs, such as regular graphs (Arvind et al., 2017).

As a consequence of this, in the seminal work *"Can Graph Neural Networks Count Substructures?"* (Chen et al., 2020), it was shown that GNNs are unable to count arbitrary subgraphs, a capability that is crucial for many real-world applications. The ability to detect and count substructures in graphs is of particular importance in fields such as chemistry and biology, where specific molecular substructures often determine functional properties.

The limited expressivity of GNNs has motivated the development of more expressive architectures, including higher-order GNNs that operate on k-tuples of nodes (Morris et al., 2019; Maron et al., 2019), subgraph GNNs, which transform the original graph into a set of modified subgraphs before applying GNN models (Cotta et al., 2021; Bevilacqua et al.,

Table 1: Test set results for *subgraph counting* with a GNN on molecular graphs. Reported: AU-ROC for multi-class classification and normalized mean avg. error of the prediction (see Sect. D.1).

| Dataset | Metric | Pattern | | | |
|---|---|---|---|---|---|
| Mutagenicity | nMAE | 0.074 | 0.043 | 0.161 | 0.167 |
| | AUC | 0.870 | 0.910 | 0.949 | 0.926 |
| MCF-7 | nMAE | 0.019 | 0.013 | 0.031 | 0.011 |
| | AUC | 0.941 | 0.859 | 0.955 | 0.925 |
| ZINC | nMAE | 0.029 | 0.009 | 0.025 | 0.009 |
| | AUC | 0.957 | 0.963 | 1.000 | 0.995 |
| ogbg-molhiv | nMAE | 0.001 | 0.004 | 0.011 | 0.002 |
| | AUC | 0.923 | 0.915 | 0.917 | 0.972 |
| ogbg-molpcba | nMAE | 0.000 | 0.000 | 0.000 | 0.000 |
| | AUC | 0.945 | 0.944 | 0.952 | 0.962 |
| Peptides-func | nMAE | 0.029 | 0.001 | 0.000 | 0.001 |
| | AUC | 0.977 | 0.936 | 0.940 | 0.882 |
| PCQM-Contact | nMAE | 0.006 | 0.000 | 0.002 | 0.001 |
| | AUC | 0.948 | 0.987 | 0.986 | 0.999 |

2021; Chen et al., 2020; Papp & Wattenhofer, 2022), and models that incorporate unique node identifiers (Sato et al., 2021; Pellizzoni et al., 2024). However, these approaches often come at a significant computational cost, limiting their practical applicability, or can show poor generalizability.

Interestingly, despite these theoretical limitations, we observe that *standard* GNNs are often able to count graph patterns with surprising accuracy across a variety of real-world datasets. For example, Table 1 reports the test set performance of a simple GNN model (Section A.1) for the subgraph counting task, for several patterns (Section D.1), across several widely used molecular datasets, using only atom types as node labels. This shows a surprisingly good performance for a task that is *in principle* unsolvable. While these experimental findings are somewhat limited in scope, they suggest that GNNs possess the capacity to approximate subgraph counts, at least for certain patterns and certain classes of target graphs. This apparent contradiction between theory and practice motivates our work to better understand the subgraph-counting capabilities of GNNs beyond worst-case scenarios, via a more nuanced analysis. Our contributions are the following:

(1) we provide conditions under which GNNs can efficiently realize functions on graphs that depend only on local substructures around the nodes (Theorem 2), including subgraph counting;

(2) we propose novel dynamic programming algorithms for restricted variants of subtree isomorphism (Theorem 4), and show that GNNs can efficiently simulate them (Theorem 5);

(3) we show that, in practice, many real-world graph datasets satisfy the sufficient conditions of point (1), and experimentally validate the claims of point (2) above.

Our work seeks to provide a theoretically-grounded explanation for the observed ability of GNNs in subgraph counting, bridging the gap between theoretical limitations and practical performance.

## 1.1 RELATED WORK

**Graph Neural Networks Expressivity** Following the influential papers Morris et al. (2019); Xu et al. (2018) that exposed the constraints of GNNs due to their expressiveness being limited by the 1-WL test (Weisfeiler & Leman, 1968), there has been a surge in research aimed at developing more capable GNNs. A notable strategy has been to create GNNs that simulate higher-order WL (Grohe, 2017) or Folklore-WL (Cai et al., 1992) tests, as demonstrated by k-GNNs (Morris et al., 2019) and k-FGNNsp (Maron et al., 2019). However, their computational and memory requirements are often impractical. Some of the subsequent approaches exploited graph locality and sparsity (Morris et al., 2022; Zhang et al., 2023; Frasca et al., 2022). An additional research direction involves subgraph GNNs (Cotta et al., 2021; Qian et al., 2022; Bevilacqua et al., 2021). Finally, individualization schemes (Pellizzoni et al., 2024; Bechler-Speicher et al., 2024) have been proposed by several works (Murphy et al., 2019; Dasoulas et al., 2020; Franks et al., 2021) to enhance the expressivity of GNNs and obtain universal function approximators (Abboud et al., 2021). Sato (2020); Morris et al. (2023b) offer a more comprehensive overview. Xu et al. (2020) demonstrated that GNNs can learn to mimic classical graph algorithms, and provided a framework for studying the complexity of simulating combinatorial algorithms with different architectures. For a comprehensive overview of GNNs' capabilities in algorithmic tasks, readers are referred to Cappart et al. (2023).

**Combinatorial subgraph counting** Subgraph (both induced and not) isomorphism, and the related counting tasks, are NP-hard (Alon et al., 1995). Due to the practical relevance of the tasks, several efficient search algorithms have been developed (Carletti et al., 2017; McCreesh et al., 2020). Another line of research, called color-coding, uses a dynamic programming approach (Alon et al., 1995; 2008; Arvind & Raman, 2002). For the task of finding all frequent subgraphs in a dataset, there exist specialized algorithms (Nijssen & Kok, 2005; Kuramochi & Karypis, 2004).

**Subgraph counting and GNNs** Chen et al. (2020) showed that GNNs are unable to count arbitrary subgraphs, and Zhang et al. (2024) obtain a full characterization of the subgraphs that can be counted on arbitrary graphs, i.e., considering the worst-case scenario. The subgraph counting problem has also been addressed on arbitrary graphs with ad-hoc architectures (Chen et al., 2020; Tahmasebi et al., 2023; Huang et al., 2023b; Paolino et al., 2024), with positional encodings (Huang et al., 2023a) or with random node features (Kanatsoulis & Ribeiro, 2024). In this work, we instead focus on standard message-passing GNNs. Subgraphs have also been used to improve the expressivity of

GNNs by using subgraph counts as features (Bouritsas et al., 2022) or by extending message-passing to subgraphs (Wang et al., 2023).

## 2 PRELIMINARIES

In what follows, we define a graph as a tuple $G = (V_G, E_G, L_G)$, with $V_G$ a finite set of nodes, and $E_G \subseteq \{\{u, v\} : u \neq v \in V_G\}$ a set of undirected edges. We define the vertex-label function as $L_G : V_G \to \Sigma$, with a finite set of labels $\Sigma$. For the sake of simplicity, we consider edges to be unlabeled. We define the neighborhood of a node as $\mathcal{N}(v) = \{w \in V_G : \{v, w\} \in E_G\}$. We say that two graphs $G$ and $H$ are isomorphic, denoted as $G \simeq H$, if there exists a bijective mapping $\pi : V_G \to V_H$, called isomorphism, such that $L_G(v) = L_H(\pi(v))$, $\forall v \in V_G$ and $\{\pi(u), \pi(v)\} \in E_H$ if and only if $\{u, v\} \in E_G$. The isomorphism relation induces equivalence classes, which we call, with abuse of notation, graphs. The group of isomorphisms from $G$ to itself is called the automorphism group $\mathrm{Aut}(G)$. A *subgraph isomorphism* from $G$ into $H$ is an injective mapping $\pi : V_G \to V_H$ s.t. $\{\pi(u), \pi(v)\} \in E_H$ for every $\{u, v\} \in E_G$ and $L_G(v) = L_H(\pi(v))$, $\forall v \in V_G$. We call it an induced subgraph isomorphism if furthermore for all pairs $u, v \in G$, if $\{\pi(u), \pi(v)\} \in E_H$ then $\{u, v\} \in E_G$. With *counting*, we denote the task of counting such maps from a pattern $P$.

We denote sets of graphs by $\mathcal{G}$. Moreover, given a graph $G$ and nodes $u, v \in V_G$, we say that they belong to the same orbit if $\exists \pi \in \mathrm{Aut}(G)$ such that $\pi(u) = v$, and denote it with $(G, u) \simeq (G, v)$. We denote the set of orbits on a set of graphs $\mathcal{G}$ with $\mathcal{V}_{\mathcal{G}} = \{(G, u) : G \in \mathcal{G}, v \in V_G\}/\simeq$. Given two nodes $u, v \in V_G$, we define with $d_G(u, v)$ their shortest-path distance in $G$. A tree is a graph with no cycles. We denote with $T_r$ the tree rooted in $r \in V_T$, and define recursively $\mathrm{children}(r) = \mathcal{N}(r)$ and if $q \in \mathrm{children}(p)$ then $\mathrm{children}(q) = \mathcal{N}(q) \setminus \{p\}$. We define the height of the tree as $\max_{p \in V_T} d_T(r, p)$ and the truncated tree $T_r^\ell$ as the subgraph of $T_r$ induced by the nodes $p$ such that $d_T(r, p) \leq \ell$. An egonet $\mathrm{EGO}_u^k(G)$ of a node $u \in V_G$ is defined as the induced subgraph of $G$ on $\{v \in V_G : d_G(u, v) \leq k\}$, with $u$ marked with a dedicated "root" label.

**The Weisfeiler–Leman algorithm** The color refinement algorithm, also known as 1-Weisfeiler–Leman (denoted as WL), is a heuristic algorithm for the graph isomorphism problem. Let $\mathrm{WL}^0(G, v) = L_G(v) \in \mathbb{N}$ be the initial color of node $v \in V_G$. Then the algorithm updates vertex colors as $\mathrm{WL}^\ell(G, v) = \mathrm{HASH}\left(\mathrm{WL}^{\ell-1}(G, v), \{\!\{\mathrm{WL}^{\ell-1}(G, w) : w \in \mathcal{N}(v)\}\!\}\right) \in \mathbb{N}$, with HASH an injective map, at iteration $\ell > 0$. Two graphs are deemed $\ell$-hop WL-isomorphic, denoted as $G \simeq_{\mathrm{WL}_\ell} H$, if $\{\!\{\mathrm{WL}^\ell(G, v) : v \in V_G\}\!\} = \{\!\{\mathrm{WL}^\ell(H, v) : v \in V_H\}\!\}$, and WL-isomorphic if it holds for $\ell = |V_G|$, denoted as $G \simeq_{\mathrm{WL}} H$. Note that $G \simeq H \implies G \simeq_{\mathrm{WL}} H$, but the converse is not true. We call a graph $G$ *WL-amenable* if $\forall H$ such that $G \not\simeq H$, $G \not\simeq_{\mathrm{WL}} H$.

**Coverings of graphs** For graphs $H$ and $G$, the mapping $\phi : V_H \to V_G$ is called a *homomorphism* if it preserves all edges and labels, i.e., if $\{\phi(u), \phi(v)\} \in E_G$ for all $\{u, v\} \in E_H$ and $L(v) = L(\phi(v))$ for all $v \in V_H$. A homomorphism $\phi$ is called locally *locally injective* or *bijective* if for every node $v \in V_H$, the mapping $\phi_v : \mathcal{N}(v) \to \mathcal{N}(\phi(v))$ is injective, respectively bijective. If there exists a locally bijective homomorphism from a graph $H$ to a graph $G$, we say that $H$ *covers* $G$. The *universal cover* (Angluin, 1980) of $G$ given a node $u \in V_G$ is a (possibly infinite) tree, denoted $U_u(G)$, that covers any graph which covers $G$. It holds that $U_u^\ell(G) \simeq U_v^\ell(H)$ if and only if $\mathrm{WL}^\ell(G, u) = \mathrm{WL}^\ell(H, v)$ (Krebs & Verbitsky, 2015). An illustration can be found in Figure 1.

**Graph neural networks** Message passing graph neural networks (GNNs), given a graph $G$, iteratively produce for each node $v \in V_G$, at each level $\ell = 1, \ldots, \mathcal{L}$, the embeddings $h_v^\ell \in \mathbb{R}^{d_\ell}$ by taking into account *messages* coming from its neighbors $\mathcal{N}(v)$. More formally, the embedding of node $v$ is updated as $h_v^\ell = f_{\mathrm{upd}}\left(h_v^{\ell-1}, f_{\mathrm{agg}}\left(\{\!\{h_u^{\ell-1} : u \in \mathcal{N}(v)\}\!\}\right)\right)$, where $f_{\mathrm{agg}}$ and $f_{\mathrm{upd}}$ are the aggregate and the update operations, respectively. The first layer of the GNN is fed with the initial node embeddings $h_v^0$, e.g. one-hot encodings of the node labels. Finally, one can get a graph-level readout $h_G^{\mathcal{L}}$ by aggregating the output node embeddings via a function $f_{\mathrm{out}}$. In Xu et al. (2018) it was shown that there exist injective functions $f_{\mathrm{agg}}$, $f_{\mathrm{upd}}$ and $f_{\mathrm{out}}$ yielding GNNs that are provably as expressive as color refinement. We denote as $\mathrm{GNN}_\ell^{\mathrm{node}} = \{(G, v) \mapsto h_v^\ell\}$ the class of parametric node-level functions formed by such a model with $\ell$ message passing layers. Moreover, let $\mathrm{GNN}_\ell = \{G \mapsto h_G^\ell\}$ be the class of parametric graph-level functions.

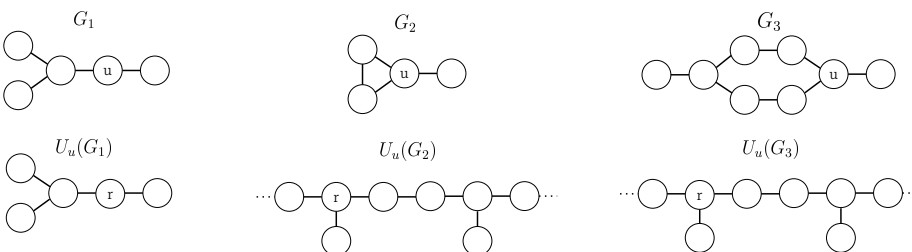

Figure 1: Three graphs and their universal covers rooted in a node $u$. Since $G_1$ is a tree, its universal cover is $G_1$ itself. The graphs $G_2$ and $G_3$ have isomorphic (infinite) universal covers. In fact, $G_3$ is a covering of $G_2$.

## 3 UNIVERSALITY ON WL-DISTINGUISHABLE GRAPHS

The classical negative result of Chen et al. (2020, Theorem 3.3) is based on pairs of WL-indistinguishable graphs such that, for a pattern $P$, one contains it as an induced subgraph while the other one does not. This result, however, requires that the set of graphs at hand features specific WL-indistinguishable graphs, a scenario that is unrealistic to happen in practice (e.g., see Table 2).

To overcome this limitation, we restrict to studying the ability of GNNs to solve the subgraph counting tasks (for a fixed pattern graph $P$) on specific sets of graphs $\mathcal{G}$, phrasing the task as a *promise problem* (Even et al., 1984). In fact, if the graphs at hand can all be distinguished by WL, we have the following positive result, which follows directly from Morris et al. (2023a).

**Proposition 1.** *Let $\mathcal{G}$ be a set of graphs such that $\forall G_1, G_2 \in \mathcal{G}$, $G_1 \not\simeq_{\mathrm{WL}} G_2$ and $|V_G| \leq n$, $\forall G \in \mathcal{G}$. Let $f : \mathcal{G} \to \mathbb{R}$ be any function. Then, there exists a function class $\mathrm{GNN}_{\ell=n}$ realized by a GNN model such that $f \in \mathrm{GNN}_{\ell=n}$.*

In particular, if the function $f$ is the (possibly induced) subgraph counting function, GNNs can count subgraphs on $\mathcal{G}$. In fact, if the set $\mathcal{G}$ is composed of WL-amenable graphs, the proposition always holds. Therefore, for several classes of graphs, such as trees and forests, which are known to be amenable (Arvind et al., 2017), as well as random graphs, which are known to be amenable with high probability (Babai et al., 1980), GNNs are indeed able to count subgraphs. This result, although it is the first step beyond the worst-case analysis of Chen et al. (2020), has several limitations.

First, the proof of universality of GNNs on WL-distinguishable graphs relies on a model with an impractical number of layers, that is able to distinguish all graphs and remember by heart the value of the function for each graph. Therefore, the size of the MLP after the graph-level pooling of the node embeddings must be (at least) linear in $|\mathcal{G}|$. In particular, $|\mathcal{G}|$ can in general grow exponentially with the maximum graph size $n$. Thus, the results hold only for sets of *bounded-size graphs*.

Secondly, and relatedly to the first problem, the model used in the proof of the proposition has high sample complexity. Informally speaking, the sample complexity of a model class is the number of training samples needed for the model to generalize well to unseen data, and is usually proportional to the number of parameters in the model. For real-valued function classes, one can characterize their sample complexity via the pseudo-dimension (Anthony & Bartlett, 1999; Mohri et al., 2018), as lower pseudo-dimension implies lower sample complexity. See Section A.2 for formal definitions. Indeed, this model has pseudo-dimension $\mathrm{Pdim}(\mathrm{GNN}_\ell) = |\mathcal{G}|$ (see Lemma 1, and Morris et al. (2023a)), implying that the model is in general incapable of generalizing to unseen data.

Therefore, while such a model can count subgraphs on the training set, it's hard to argue that it is able to *learn* to count subgraphs. In the next sections, we address this issue.

## 4 UNIVERSALITY FOR LOCAL FUNCTIONS

The loose results of the previous section are partly due to the fact that we treat subgraph counting as an arbitrarily complex function. In fact, we can exploit the fact that subgraph counting depends only on local substructures around each node.

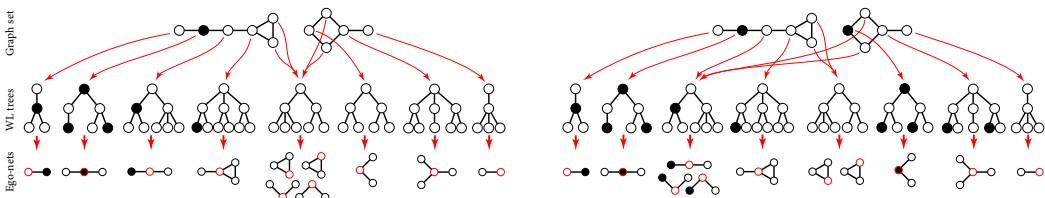

Figure 2: The set of graphs on the left is not $(2, 1)$-identifiable, while the one on the right is.

**Definition 1.** *We define a function $f : \mathcal{G} \to \mathbb{R}$ to be node-decomposable into $g$ if it can be written as $f(G) = \sum_{u \in V_G} g(G, u)$, for some function $g : \mathcal{V}_{\mathcal{G}} \to \mathbb{R}$. Moreover, $g$ is said to be $k$-local if $g(G_1, u) \neq g(G_2, v)$ implies $\mathrm{EGO}_u^k(G_1) \not\simeq \mathrm{EGO}_v^k(G_2)$. With some abuse of notation, we say that a function $f$ that is node-decomposable into $k$-local functions is also $k$-local.*

Subgraph counting and induced subgraph counting, i.e., counting the number of (induced) subgraph isomorphisms $\phi$ from $P$ to $G$, are indeed node-decomposable into *rooted* (induced) subgraph counting, which only counts, for a node $u \in V_G$, the number of (induced) subgraph isomorphisms such that $\phi(p) = u$ for some fixed $p \in V_P$ (Lemma 2). Moreover, if $P$ is the pattern at hand and there exists a node $p \in V_P$ such that $\max_{q \in V_P} d_P(p, q) \leq k$, i.e., the pattern has radius at most $k$, then these functions are $k$-local. Indeed, if $\phi$ is a (induced) subgraph isomorphism from $P$ to a subgraph of $G$ such that $\phi(p) = u$, we have that $\phi(q) \in V_{\mathrm{EGO}_u^k(G)}, \ \forall q \in V_P$.

This gives hope for a GNN model that relies only on local structures around nodes to correctly solve the subgraph counting tasks, thus having a number of parameters independent of $n$, the maximum graph size. Nonetheless, the following impossibility result shows that message passing GNNs are not able to represent all functions on node orbits, even when the graphs of the domain of such functions are all distinguishable by WL. Indeed, if two non-isomorphic graphs have isomorphic universal covers, such as $G_2$ and $G_3$ in Figure 1, then there exist nodes that are not distinguishable by a GNN. In fact, GNNs are unable to perform rooted subgraph counting on these graphs.

**Theorem 1.** *There exists a set of graphs $\mathcal{G}$ such that $\forall G_1, G_2 \in \mathcal{G}$, $G_1 \not\simeq_{\mathrm{WL}} G_2$ and a function $f : \mathcal{V}_{\mathcal{G}} \to \mathbb{R}$ for which there exists no function class $\mathrm{GNN}_\ell^{\mathrm{node}}$ realized by a GNN model such that $f \in \mathrm{GNN}_\ell^{\mathrm{node}}, \ \forall \ell$.*

In fact, having no two graphs that have isomorphic universal covers is not only a necessary condition for GNNs to represent any function on node orbits, but also a sufficient one (Krebs & Verbitsky, 2015). Nonetheless, the model might require a number of parameters that is exponential in the maximum graph size $n$ to distinguish all node orbits. In the next section, we discuss a finer grained sufficient condition for node orbits to be distinguished by a GNN model with number of layers and number of parameters independent of $n$, and show that it allows these models to learn sample efficiently the subgraph counting tasks.

### 4.1 FINE-GRAINED DISTINGUISHABILITY OF NODES

Since the subgraph counting tasks are $k$-local, we do not need GNNs to be able to distinguish any two node orbits, but rather to distinguish ego-nets of radius $k$. In general, one might need $\ell \geq k$ message passing layers to do so. Then, we can characterize the functions that can be realized by a GNN with $\ell$ layers via the notion of $(\ell, k)$-identifiability.

**Definition 2.** *Let $\mathcal{G}$ be a set of graphs. We say that $\mathcal{V}_{\mathcal{G}}$ is $(\ell, k)$-identifiable if $\forall (G_1, u), (G_2, v) \in \mathcal{V}_{\mathcal{G}}, U_u^\ell(G_1) \simeq U_v^\ell(G_2)$ implies that $\mathrm{EGO}_u^k(G_1) \simeq \mathrm{EGO}_v^k(G_2)$. If $\mathcal{V}_{\mathcal{G}}$ is $(\ell, k)$-identifiable, we also say that $\mathcal{G}$ is $(\ell, k)$-identifiable.*

Figure 2 gives a visual representation of a set of graphs that is not $(2, 1)$-identifiable, as two non-isomorphic ego-nets of radius 1 have the same truncated universal covers of depth 2, and thus the same WL color. It also depicts a set of graphs that is $(2, 1)$-identifiable. Interestingly, both sets are $(3, 1)$-identifiable. Indeed, as experimentally shown in Section 6, graph sets that are not $(\ell, k)$-identifiable are rare in practice. We define $\eta_{\mathcal{G}, \ell} = |\{U_u^\ell(G) : (u, G) \in \mathcal{V}_{\mathcal{G}}\}/{\simeq}|$, the number of truncated universal covers in $\mathcal{G}$. We then have the following results.

**Theorem 2.** *Let $\mathcal{G}$ be a $(\ell, k)$-identifiable set of graphs. Consider any $k$-local function $f : \mathcal{V}_{\mathcal{G}} \to \mathbb{R}$. Then, there exists a function class $\mathrm{GNN}_{\ell, \mathcal{G}}^{\mathrm{node}}$ realized by a GNN model with $O(\eta_{\ell, \mathcal{G}}^2 \cdot \ell)$ parameters and $\ell$ layers such that $f \in \mathrm{GNN}_{\ell}^{\mathrm{node}}$.*

**Corollary 1.** *Let $\mathcal{G}$ be a $(\ell, k)$-identifiable set of graphs. Let $\mathrm{GNN}_{\ell}$ be a function class realized by a GNN model with sum-aggregation $f_{\mathrm{out}}(\{\!\{h_u^{\ell} : u \in V_G\}\!\}) = \sum_{u \in V_G} h_u^{\ell}$. Then $\mathrm{GNN}_{\ell}$ can perform both subgraph counting and induced subgraph counting of patterns of radius at most $k$.*

Notably, the GNN model that realizes Corollary 1 has number of parameters that is independent of the maximum graph size $n$, depending only on $\ell$ and $\eta_{\mathcal{G}, \ell}$, and has a simple readout function. This allows not only to count arbitrary (small) subgraphs on sets of graphs with unrestricted maximum graph size, but also to learn such tasks sample efficiently, i.e., with fewer training samples.

**Theorem 3.** *The function class $\mathrm{GNN}_{\ell}$ of Cor. 1 has pseudo-dimension $\mathrm{Pdim}(\mathrm{GNN}_{\ell}) \leq \eta_{\ell, \mathcal{G}} + 1$.*

As implied by the previous theorem, the sample complexity of the simple model that realizes Corollary 1 can be bounded based solely on the number of local structures around nodes, and independently on global properties of the graphs.

## 5 ALGORITHMICALLY-ALIGNED GNNS FOR TREE PATTERNS

The positive results in previous sections solve subgraph counting by recognizing entire graphs or ego-nets around nodes. This approach could solve tasks beyond subgraph counting, potentially producing unnecessarily large models. We now explore the ability of GNNs to simulate combinatorial algorithms for subgraph counting-related tasks, as it has been noted in the literature that GNNs align well with dynamic programming (DP) algorithms (Xu et al., 2020; Nerem et al., 2025). We develop novel algorithms solving the (non-induced) subgraph isomorphism problem on restricted pattern and target graph classes. We demonstrate GNNs can efficiently simulate these algorithms, situating the resulting models within the algorithmic alignment framework (Xu et al., 2020).

Our algorithms are inspired by the *color coding* algorithm (Alon et al., 1995; 2008), which finds subgraph isomorphisms $\phi$ from the pattern tree $T$ to the target graph $G$ such that $c(\phi(p)) \neq c(\phi(q)), \forall p \neq q \in V_T$, where $c(u)$ represents a color assigned to node $u$. Enforcing that the images of the pattern graph's nodes have different colors ensures injectivity. In the original algorithm, this condition is enforced by assigning random colors to the nodes of $G$, and repeating the procedure multiple times to boost the success probability. This assignment of random colors is however impossible to simulate with message passing. Therefore, we will color the target nodes based on WL colors, which can be obtained with message passing layers. This however requires to modify the color coding algorithm to relax the condition on the target graphs' colors, as it is unrealistic to assume that any subgraph in $G$ matching the pattern has all nodes belonging to different WL classes.

### 5.1 A DYNAMIC PROGRAM FOR COLORFUL SUBTREE ISOMORPHISM

We tackle the *subgraph isomorphism* problem from tree patterns. Say that the tree pattern at hand is $T$, rooted in $r$ and with nodes endowed with labels $L : V_T \to \Sigma$. We denote with $T_p$ the subtree of $T_r$ rooted in $p \in V_T$, i.e. the subgraph induced by the descendants of $p$. The nodes of the target graph $G$ are endowed with labels $L : V_G \to \Sigma$ and colors $c : V_G \to \Omega$ such that $L(u) \neq L(v) \implies c(u) \neq c(v)$.

**High-level description:** Our dynamic program TREE-COLSI determines if a subgraph isomorphism $\phi$ from $T_p$ to $G$ exists, where $\phi(p) = u$, for nodes $p \in V_T$ and $u \in V_G$. For a leaf $p$, we check if $u$ has the same label. For internal $p$ with matching label to $u$, we build $\phi$ by mapping $p$ to $u$ and its children to distinct neighbors of $u$, using inductively gathered information from $\mathcal{N}(u)$. Indeed, if there exist subgraph isomorphisms $\phi_q$ from all the subtrees $T_q$, for $q \in \mathrm{children}(p)$, mapping each $q$ to distinct neighbors of $u$, one can create a homomorphism from $T_p$ to $G$ by mapping $p$ to $u$ and the nodes in the $T_q$'s to their images in $\phi_q$.

However, the resulting map could be not injective. Indeed, it could happen, e.g, that in the maps $\phi_{q_1}, \phi_{q_2}$ a child of $q_1$ and a child of $q_2$ are mapped to the same target graph's node. To avoid this, for each discovered map $\phi_q$ from $T_q$ to $G$ with $\phi_q(q) = v$, we store a colorset $C_{\phi_q}$ with the colors of the images of the descendants of $q$ in a set $\mathcal{C}_{v, q}$. Then, when we create the map $\phi$ such that $\phi(p) = u$ by

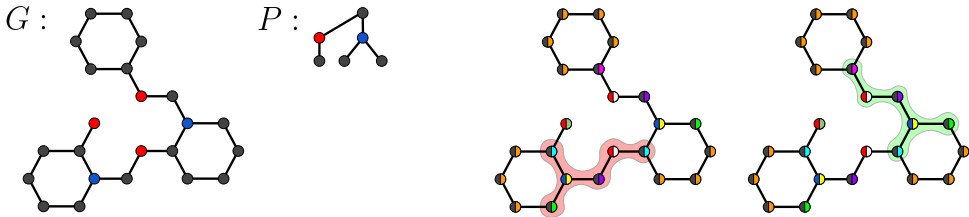

Figure 3: A graph $G$ and a tree pattern $P$. The copies of $G$ on the right have colors (right half of the nodes) obtained from one WL iteration. The subgraph isomorphism from $P$ to the red-highlighted subgraph of $G$ is not quite-colorful, while the one to the green-highlighted subgraph of $G$ is.

merging the maps $\phi_q$ from all the nodes $q \in \text{children}(p)$ to distinct neighbors of $u$, we require that the images of any two pattern graph's nodes that could be mapped to the same target graph's node have different colors. This will allow to obtain injectivity.

**Formal description:** Let $h$ be the height of the tree $T_r$ and $T^h = \{r\}$. Let $T^{\ell-1} = \{q \in \text{children}(p) : p \in T^\ell\}, \ell = 1, \ldots, h$. Clearly, $T^0$ consists of only leaves of the tree. The dynamic program proceeds in layers, processing at the same time all the pattern graph's nodes in $T^\ell$. Therefore, the output of $\text{TREE-COLSI}_c(u, \ell)$ will be a dictionary mapping nodes $p \in T^\ell$ to sets $\mathcal{C}_{u,p}$. In particular, for nodes $u \in V_G$, $p \in V_T$, $\mathcal{C}_{u,p}$ is a set of *colorsets* $C \in 2^\Omega$. We have that if $C_\phi \in \mathcal{C}_{u,p}$, then there exists a subgraph isomorphism $\phi$ from $T_p$ to $G$ mapping $p$ to $u$.

---

**Algorithm 1:** $\text{TREE-COLSI}_c(u, \ell)$

1   let $\mathcal{C}_{u,\ell}$ be an empty dictionary
2   **if** $\ell > 0$
3     **for** $v \in \mathcal{N}(u)$ **do**
4       $\mathcal{C}_{v,\ell-1} = \text{TREE-COLSI}_c(v, \ell-1)$
5       $\mathcal{C}_{v,q} = \mathcal{C}_{v,\ell-1}[q], \ \forall q \in \text{children}(p), \forall p \in T^\ell$
6   **for** $p \in T^\ell$ **do**
7     **if** $\text{children}(p) == \emptyset$
8       **if** $L(u) \neq L(p)$: $\mathcal{C}_{u,\ell}[p] = \emptyset$
9       **else**: $\mathcal{C}_{u,\ell}[p] = \{\emptyset\}$
10       **continue**
11     **if** $L(u) \neq L(p)$: set $\mathcal{C}_{u,\ell}[p] = \emptyset$ and **continue**
12     let $\mathcal{C}_{u,p} = \emptyset$ and $(q_1, \ldots, q_\delta) = \text{children}(p)$
13     **for** seq. of distinct nodes $(v_1, \ldots, v_\delta)$ from $\mathcal{N}(u)$ **do**
14       **if** $\exists i : \mathcal{C}_{v_i,q_i} == \emptyset$: **continue**
15       let $\{C_{v_i,j} : j = 1, \ldots, |\mathcal{C}_{v_i,q_i}|\} = \mathcal{C}_{v_i,q_i}$
16       **for** seq. of colorsets $(C_{v_1,j_1}, \ldots, C_{v_\delta,j_\delta})$ **do**
17         $C = \text{MERGE}((C_{v_1,j_1}, \ldots, C_{v_\delta,j_\delta}))$
18         **if** $C \neq \epsilon$
19           $\mathcal{C}_{u,p} = \mathcal{C}_{u,p} \cup \{C\}$
20     $\mathcal{C}_{u,\ell}[p] = \mathcal{C}_{u,p}$
21   **return** $\mathcal{C}_{u,\ell}$

---

**Algorithm 2:** $\text{MERGE}((C_1, \ldots, C_\delta))$

1   flag $= (c(u) \notin C_i, \forall i)$
2   flag $=$ flag and $(c(v_i) \notin C_j, \forall i \neq j)$
3   flag $=$ flag and $(C_i \cap C_j == \emptyset \ \forall i \neq j)$
4   **if** flag $==$ false: **return** $\epsilon$
5   let $\bar{C}_i = C_i \cup \{c(v_i)\}, \ \forall i$
6   let $C = \bigcup_i \bar{C}_i$
7   **return** $C$

---

The dynamic program $\text{TREE-COLSI}_c$, applied with colors $c$, is described in Algorithm 1. It takes as input a target graph node $u$ and a level $\ell$. It returns a dictionary mapping from pattern graph nodes to sets of colorsets. Lines 2-5 gather information from the neighbors of $u$ by recursively calling the procedure on the nodes in $T^{\ell-1}$, if $\ell > 0$. The for-loop of line 6 iterates over the nodes $p$ in $T^\ell$ to compute the sets $\mathcal{C}_{u,p}$. Lines 7-10 handle the case when $p$ is a leaf. For an internal node $p$, if $L(p) = L(u)$, the for-loop on line 13 tries all possible assignments of the nodes in $\text{children}(p)$ to the nodes in $\mathcal{N}(u)$. The loop on line 16 tries, for each such assignment, all the combinations of colorsets belonging to the sets $\mathcal{C}_{v_1,q_1}, \ldots, \mathcal{C}_{v_\delta,q_\delta}$. For one such sequence $C_1, \ldots, C_\delta$, the algorithm checks with $\text{MERGE}$ if the recursively-obtained sub-maps can be merged to create a map mapping $p$ to $u$, by checking injectivity via the colors of the image of the map. A dummy value $\epsilon$ is used to signal that the map cannot be created. If the map can be created, the algorithm builds a colorset $C_{(v_1,\ldots,v_\delta),(j_1,\ldots,j_\delta)}$ associated with the map, which is inserted in $\mathcal{C}_{u,p}$. One would call the algorithm with $\text{TREE-COLSI}_c(u, h)$, with $h$ the height of the tree pattern, and would obtain the dictionary $\mathcal{C}_{u,h}$. One can then access $\mathcal{C}_{u,r}$, with $r$ the root of the tree, as $\mathcal{C}_{u,h}[r]$. An example of the execution of the algorithm can be found in Section A.4

Due to the "non-allowed colors" strategy for enforcing injectivity, the dynamic program will not detect the subgraph isomorphisms from $T$ to $G$ where some specific pairs of pattern graph nodes are mapped to target graph nodes with the same color. We now formally characterize the family of maps that can be recognized.

**Definition 3.** *Let $T_r$ be a tree, $G$ be a graph whose nodes are endowed with colors $c : V_G \to \Omega$, and $\phi : V_{T_r} \to V_G$ a map. We say $\phi$ is quite-colorful (w.r.t. c) if (i) $\forall p \in V_{T_r}$, $\forall q \in \mathrm{children}(p)$ and $\forall t \in \mathrm{children}(q)$ it holds that $c(\phi(p)) \neq c(\phi(t))$, and (ii) $\forall p, q \in V_{T_r}$ such that $d_{T_r}(p, q) \geq 3$ it holds that $c(\phi(p)) \neq c(\phi(q))$.*

**Theorem 4.** *Let $T_r$ be a tree of height $h$, $G$ be a graph whose nodes are endowed with colors $c : V_G \to \Omega$. Then, $\mathrm{TREE\text{-}COLSI}_c(u, h)[r] \neq \emptyset$ if and only if there is a quite-colorful subgraph isomorphism $\phi$ from $T_r$ to $G$ such that $\phi(r) = u$.*

A visual representation for a subgraph isomorphism map that is not quite-colorful and for one that is quite-colorful can be found in Figure 3. A simple sufficient (but not necessary) condition to ensure that the algorithm solves subgraph isomorphism is that labels of the pattern graph itself respect the conditions for quite-colorfulness, i.e., $\forall p \in V_{T_r}$, $\forall q \in \mathrm{children}(p)$ and $\forall t \in \mathrm{children}(q)$ it holds that $L(\phi(p)) \neq L(\phi(t))$, and $\forall p, q \in V_{T_r}$ such that $d_{T_r}(p, q) \geq 3$ it holds that $L(\phi(p)) \neq L(\phi(q))$, as the map will be guaranteed to be parent-colorful. We call these quite-colorful patterns.

The algorithm is designed, rather than to minimize the computational complexity (Section A.3), to align with message passing, as mapping $p$ to $u$ only requires information from $u$'s neighbors. This allows GNNs to simulate the algorithm, as we show in the next section.

## 5.2 GRAPH NEURAL NETWORKS CAN SIMULATE TREE-COLSI

We show that a GNN model can simulate the execution of $\mathrm{TREE\text{-}COLSI}_c$, with the colors $c$ of the target nodes being WL colors, i.e., such that $c(u) = c(v)$ iff $U_v^l(G) \simeq U_u^l(G)$. Note that such WL colors can be obtained by message passing on both labeled and unlabeled graphs.

The structure of the dynamic program is, by design, aligned with the message passing framework used by GNNs. Indeed, in the dynamic program, the computation of a set $\mathcal{C}_{u,\ell}$ depends uniquely on the sets $\mathcal{C}_{v,\ell-1}$ for all $v \in \mathcal{N}(u)$, as well as the colors $c(v)$ for $v \in \mathcal{N}(u) \cup \{u\}$. Indeed, while also the labels $L(u)$ are used, these are uniquely identified by the colors $c(u)$ by construction.

Therefore, one can encode the information $(\mathcal{C}_{v,\ell-1}, c(v))$ in the node embedding $h_v^{\ell-1}$ and communicate this information to the neighbors of $v$. Special care needs to be taken in order for $f_{\mathrm{agg}}\left(\{\!\{h_v^{\ell-1} : v \in \mathcal{N}(u)\}\!\}\right)$ to uniquely represent the multiset $\{\!\{(\mathcal{C}_{v,\ell-1}, c(v)) : v \in \mathcal{N}(u)\}\!\}$. The function mapping $(\mathcal{C}_{u,\ell-1}, c(u)), \{\!\{(\mathcal{C}_{v,\ell-1}, c(v)) : v \in \mathcal{N}(u)\}\!\}$ to $(\mathcal{C}_{u,\ell}, c(u))$ can then be realized by a MLP in the $f_{\mathrm{upd}}$ function, which therefore only simulates lines 6-21 of the dynamic program. Because of this, the GNN algorithmically aligns (Xu et al., 2020, Definition 3.4) with the execution of the dynamic program, which can lead to better generalization (Xu et al., 2020, Theorem 3.6). Moreover, the MLP will have a number of parameters that depends on the maximum number of distinct elements $(\mathcal{C}_{u,\ell-1}, c(u)), \{\!\{(\mathcal{C}_{v,\ell-1}, c(v)) : v \in \mathcal{N}(u)\}\!\}$ over all $u \in \mathcal{V}_\mathcal{G}$ and $\ell = 1, \ldots, h$, which we call $\zeta_{l,T,\mathcal{G}}$. Then, we have that a GNN can efficiently simulate the DP, and therefore we can apply to GNNs the results of Theorem 4.

**Theorem 5.** *Let $\mathcal{G}$ be a set of graphs of bounded degree, $T_r$ be a tree of height $h$. Let $f(G) = 1$ if $\exists u \in V_G : \mathrm{TREE\text{-}COLSI}_c(u, h)[r] \neq \emptyset$ and 0 otherwise. Then, there exists a function class $\mathrm{GNN}_{l+h}$ realized by a GNN model with $l + h$ layers and $O\left(\eta_{l,\mathcal{G}}^2 \cdot l + \zeta_{l,T_r,\mathcal{G}} \cdot h\right)$ parameters such that $f \in \mathrm{GNN}_{l+h}$.*

We prove in Lemma 5 that the quantity $\zeta_{l,T,\mathcal{G}}$, which plays a crucial role in the complexity of the model, can be upper bounded by $\eta_{l+h,\mathcal{G}}$, recovering a dependency similar to the result of Theorem 2. We show there that the quantity can also be bounded based on the maximum degree of the graph $\Delta$, the size of the pattern $\kappa$ and the number of used colors $\eta_{\mathcal{G},l}$ as $O\left(\eta_{\mathcal{G},l}^{\Delta+1}/\Delta! \cdot 2^{(\Delta+1)\eta_{\mathcal{G},l}^\kappa/(\kappa-1)!}\right)$. However, when in practice there are few distinct elements $(\mathcal{C}_{u,\ell-1}, c(u)), \{\!\{(\mathcal{C}_{v,\ell-1}, c(v)) : v \in \mathcal{N}(u)\}\!\}$, e.g. if the subtrees of $T$ appear rarely in the data or appear on target nodes with few distinct colors, we get better model size bounds, which then leads to better sample complexity.

**Comparison with previous work** The result of Theorem 4 yields a generalization of Chen et al. (2020, Theorem 3.5), as maps from stars are always quite-colorful. Moreover, Zhang et al. (2024, Theorem 4.5) proves that GNNs can count subgraphs (on general graphs) if and only if the *spasm* of the pattern at hand is composed of only trees. Indeed, the spasm of a quite-colorful tree is composed of only trees. On the other hand, if a tree pattern has two nodes at distance at least 3 with the same

Table 2: Number $|\mathcal{G}/\simeq_{WL_\ell}|$ of WL isomorphism classes after $\ell$ iterations and number $|\mathcal{G}/\simeq|$ of actual isomorphism classes on common molecular datasets. Ratio between the two in brackets.

| Dataset | $|\mathcal{G}/\simeq_{WL_\ell}|$ | | | | | $|\mathcal{G}/\simeq|$ |
|---|---|---|---|---|---|---|
| | $\ell = 1$ | $\ell = 2$ | $\ell = 3$ | $\ell = 4$ | $\ell = \infty$ | |
| Mutagenicity (Kersting et al., 2016) | 3634 (0.839) | 4274 (0.985) | 4333 (0.999) | 4337 (1.000) | 4337 (1.000) | 4337 |
| MCF-7 (Kersting et al., 2016) | 26058 (0.946) | 27368 (0.994) | 27525 (0.999) | 27532 (1.000) | 27538 (1.000) | 27538 |
| ZINC (Gómez-Bombarelli et al., 2018) | 11988 (1.000) | 11994 (1.000) | 11994 (1.000) | 11994 (1.000) | 11994 (1.000) | 11994 |
| ogbg-molhiv (Hu et al., 2021; Wu et al., 2018) | 38765 (0.942) | 40942 (0.996) | 41082 (0.999) | 41102 (1.000) | 41122 (1.000) | 41122 |
| ogbg-molpcba (Hu et al., 2021; Wu et al., 2018) | 375291 (0.873) | 428220 (0.996) | 429584 (0.999) | 429730 (1.000) | 429802 (1.000) | 429802 |
| Peptides-func (Dwivedi et al., 2022; Singh et al., 2015) | 14162 (0.937) | 14679 (0.972) | 14854 (0.983) | 15075 (0.997) | 15117 (1.000) | 15117 |
| PCQM-Contact (Dwivedi et al., 2022) | 428356 (0.819) | 521598 (0.998) | 522786 (0.999) | 522836 (1.000) | 522837 (1.000) | 522837 |

color, and thus is not quite-colorful, it will have a cyclic graph in its spasm, and cannot therefore be counted by a GNN on general target graphs. Our Theorem 4 therefore helps understand the results of Zhang et al. (2024, Theorem 4.5), by providing upper bounds on the number of layers, the parameter count and the sample complexity of a GNN that is indeed able to count such substructures.

Our results however, go beyond this. Indeed, even when the pattern at hand is not quite-colorful (e.g., for an unlabeled pattern), the algorithm can exploit the colors of the target graph to elude the worst-case analysis of Zhang et al. (2024, Theorem 4.5). Indeed, given that the target graphs have enough asymmetries, the colors $c$ obtained by $l$ iterations of color refinement (or GNN message passing layers) are enough to make the subgraph isomorphism maps quite-colorful. This, as we observe in Section 6, indeed holds in practice on several real-world datasets. Note that, on adversarial examples like regular graphs, quite-colorfulness cannot be obtained for any value of $l$.

## 5.3 EXTENSIONS

We obtained a message-passing-like dynamic program for *quite-colorful* subgraph isomorphism from tree patterns. However, the quite-colorful condition can be too restrictive in some scenarios. A similar problem is the locally injective homomorphism problem from tree patterns. In this relaxation of the subgraph isomorphism problem, the map is required to be injective only on the neighborhood of each pattern graph's node. In Section B.1 we propose TREE-COLLIH, a slightly modified algorithm that solves the locally injective homomorphism problem under a weaker condition with respect to quite-colorfulness.

The algorithms described in the previous sections, as the original algorithms (Alon et al., 1995), can only deal with tree patterns. This is necessary due to the fact that the maps built on different subtrees can be merged recursively without conflicts. In fact, in Section B.2 we discuss how to extend the approach, which can still be simulated by a GNN, to cyclic patterns.

Moreover, our algorithms are described in the decision problem variant. As argued in a follow-up to the original color-coding algorithm Alon et al. (2008), one can modify the dynamic program, and the associated GNNs, to count maps. We describe how to do so in Section B.3.

## 6 EXPERIMENTAL EVALUATION

### 6.1 THE CONDITIONS FOR SUBGRAPH COUNTING HOLD IN PRACTICE

We study whether the sufficient conditions for subgraph counting that we have identified hold in practice. As in Table 1, we focus on molecular graphs, where subgraph counting has been the focus of extensive research, due to the relevance of subgraphs corresponding to functional groups. Table 2 reports the number $|\mathcal{G}/\simeq_{WL_\ell}|$ of WL isomorphism classes after $\ell = 1, 2, 3, 4, \infty$ iterations. If one compares this figure to the number $|\mathcal{G}/\simeq|$ of actual isomorphism classes, it is immediate to see that the number of WL-indistinguishable graphs is negligible in practice, as observed in Zopf (2022).

Moreover, Table 3 reports the fraction of the nodes in the dataset that are $(\ell, k)$-identifiable, namely such that their universal cover of depth $\ell$ uniquely identifies their ego-net of radius $k$. We also report the fraction of graphs for which all nodes are $(\ell, k)$-identifiable. We can observe that already for $\ell = k + 2$, more than $97\%$ of the ego-nets centered around nodes can be identified by the WL color of the node after $\ell$ iterations. This result justifies the real-world applicability of Theorem 2.

Table 3: Fraction of nodes in the dataset that are $(\ell, k)$-identifiable. In brackets, fraction of graphs in the dataset for which all nodes are $(\ell, k)$-identifiable.

| Dataset | $k = 1$ | | | | $k = 2$ | | | | $k = 3$ | | | |
|---|---|---|---|---|---|---|---|---|---|---|---|---|
| | $\ell = 1$ | $\ell = 2$ | $\ell = 3$ | $\ell = 4$ | $\ell = 2$ | $\ell = 3$ | $\ell = 4$ | $\ell = 5$ | $\ell = 3$ | $\ell = 4$ | $\ell = 5$ | $\ell = 6$ |
| Mutagenicity | 0.880 (0.265) | 0.978 (0.766) | 0.997 (0.973) | 0.999 (0.994) | 0.647 (0.071) | 0.945 (0.634) | 0.992 (0.932) | 0.997 (0.994) | 0.663 (0.120) | 0.952 (0.673) | 0.994 (0.949) | 0.997 (0.993) |
| MCF-7 | 0.389 (0.002) | 0.951 (0.619) | 0.996 (0.960) | 0.999 (0.993) | 0.323 (0.004) | 0.913 (0.308) | 0.995 (0.937) | 0.999 (0.994) | 0.615 (0.029) | 0.953 (0.502) | 0.996 (0.954) | 0.999 (0.995) |
| ZINC | 0.844 (0.116) | 0.982 (0.825) | 0.996 (0.934) | 1.000 (0.998) | 0.548 (0.003) | 0.952 (0.542) | 0.995 (0.934) | 1.000 (0.993) | 0.782 (0.054) | 0.981 (0.781) | 0.998 (0.970) | 1.000 (0.998) |
| ogbg-molhiv | 0.734 (0.073) | 0.961 (0.708) | 0.998 (0.979) | 1.000 (0.998) | 0.477 (0.013) | 0.952 (0.598) | 0.996 (0.945) | 0.999 (0.996) | 0.739 (0.078) | 0.968 (0.669) | 0.996 (0.963) | 0.999 (0.995) |
| ogbg-molpcba | 0.780 (0.036) | 0.959 (0.639) | 0.992 (0.905) | 0.999 (0.987) | 0.504 (0.002) | 0.933 (0.467) | 0.989 (0.861) | 0.998 (0.963) | 0.674 (0.025) | 0.948 (0.556) | 0.988 (0.861) | 0.998 (0.959) |
| Peptides-func | 0.825 (0.000) | 1.000 (1.000) | 1.000 (1.000) | 1.000 (1.000) | 0.825 (0.001) | 0.999 (0.960) | 1.000 (1.000) | 1.000 (1.000) | 0.711 (0.143) | 0.961 (0.551) | 0.993 (0.551) | 0.999 (0.907) |
| PCQM-Contact | 0.699 (0.001) | 0.942 (0.421) | 0.991 (0.870) | 0.997 (0.966) | 0.351 (0.001) | 0.854 (0.328) | 0.973 (0.805) | 0.995 (0.949) | 0.391 (0.006) | 0.848 (0.373) | 0.975 (0.841) | 0.998 (0.973) |

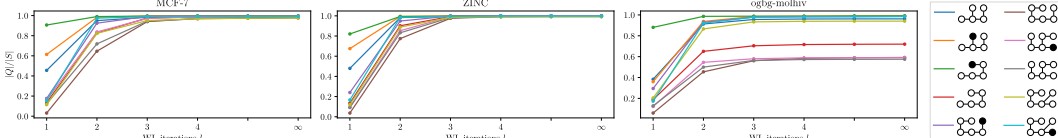

Figure 4: Ratio of the number $|Q|$ of subgraph isomorphisms that are quite-colorful to the total number $|S|$ of subgraph isomorphisms, reported for increasing numbers of WL iterations $l$ and for several (non quite-colorful) patterns.

Finally, Figure 4 reports the proportion of subgraph isomorphisms that are quite-colorful, for increasing numbers of WL iterations $l$ and for several (non quite-colorful) patterns. Full results are reported in Section D.3. The results show that for real-world datasets, such as MCF-7 and ZINC, nearly all subgraph isomorphisms are quite-colorful, already when target graphs are node-colored using only $l = 3$ iterations of color refinement. In general, we notice that, for the vast majority of the patterns, most subgraph isomorphism maps are indeed quite-colorful already for $l = 2$. There are specific patterns for which not all maps are quite-colorful, no matter the choice of $l$. Understanding the mechanisms for which these patterns can be counted, at least approximately, remains an open question for future work.

## 6.2 ADDITIONAL EXPERIMENTAL RESULTS

In Section D we report additional experimental results. In particular, we report an extended version of Table 1 with several more tree and cyclic patterns, showing that indeed GNNs can count quite well several substructures. Moreover, we validate the ability of GNNs to count quite-colorful patterns on challenging synthetic datasets, further proving the results of Section 5.

## 7 DISCUSSION AND CONCLUSIONS

Our work provides a theoretically-grounded explanation for the observed effectiveness of message passing graph neural networks (GNNs) in subgraph counting tasks, bridging the gap between theoretical limitations and practical performance. Indeed, by moving beyond worst-case scenarios, we provide a more nuanced analysis of the subgraph-counting abilities of GNNs.

In particular, we derived sufficient conditions under which GNNs can efficiently realize functions on graphs that depend only on local substructures around nodes, such as subgraph counting, and we have shown that they often hold in practice. Moreover, we developed novel algorithms for subtree isomorphism and demonstrated that GNNs can efficiently simulate them, providing a new perspective on the computational capabilities of GNNs in relation to classical graph algorithms.

Finally, we show that, in practice, more expressivity in GNN architectures is almost never needed, as in many graph datasets there are no pairs of non-isomorphic graphs that cannot be distinguished by GNNs. Therefore, having ruled out expressivity, an interesting research avenue is to investigate the true reasons (e.g. lower sample complexity, or more ease in the optimization) why several GNN architectures designed to go beyond the limitations of the 1-WL test (Morris et al., 2019) often show better performance than their simpler counterparts in several prediction tasks.

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

# A  ADDITIONAL DETAILS

## A.1  EXPRESSIVE GRAPH NEURAL NETWORKS

An example of functions $f_{\mathrm{agg}}$, $f_{\mathrm{upd}}$ and $f_{\mathrm{out}}$ that leads to models that are provably as expressive as color refinement (Morris et al., 2019), denoting $\|$ as concatenation, is

$$h_v^\ell = \mathrm{mlp}\Big(h_v^{\ell-1}\Big\| \sum_{u \in \mathcal{N}(v)} h_u^{\ell-1}\Big) \in \mathbb{R}^{d_\ell} \qquad h_G = \mathrm{mlp}\Big(\sum_{v \in V_G} h_v^l\Big) \in \mathbb{R}.$$

## A.2  SAMPLE COMPLEXITY AND PSEUDO-DIMENSION

Let $\ell(f(x), y)$ be a loss. Learning theory (Mohri et al., 2018; Shalev-Shwartz & Ben-David, 2014) is concerned with the task to bound the difference between the true risk $R(f)$ and the empirical risk $\hat{R}(f)$:

$$R_\ell(f) = \mathbb{E}_{(x,y)\sim\mathcal{D}}[\ell(f(x), y)] \quad \text{and} \quad \hat{R}_{D,\ell}(f) = \frac{1}{m}\sum_{i=1}^m \ell(f(x_i), y_i),$$

where $D = \{(x_i, y_i)\}_i \sim \mathcal{D}^m$ represents a training dataset of size $m$ sampled i.i.d. from the data generating distribution $\mathcal{D}$.

In particular, given a class of functions $\mathcal{F}$, we say that it has the *uniform convergence* property (Shalev-Shwartz & Ben-David, 2014) if there exists a function $m_{\mathcal{F}}(\epsilon, \delta)$ such that, for every $\epsilon, \delta \in \,]0, 1[$ and every distribution $\mathcal{D}$, drawing a dataset $D$ of $m \geq m_{\mathcal{F}}(\epsilon, \delta)$ i.i.d. samples from $\mathcal{D}$ yields that, with probability at least $1 - \delta$ over the choices of the samples, $\sup_{f\in\mathcal{F}} |\hat{R}_{D,\ell}(f) - R_\ell(f)| \leq \epsilon$. The function $m_{\mathcal{F}}(\epsilon, \delta)$ is called the *sample complexity* of the function class.

In general, a model class that has low sample complexity needs fewer training samples in order to generalize to unseen data. For models outputting real values, their sample complexity can be characterizes via the pseudo dimension (Anthony & Bartlett, 1999; Mohri et al., 2018).

**Definition 4.** *Let $X$ be a set and $\mathcal{F} \subseteq \ f : X \to \mathbb{R}$ a class of real-valued functions. We say that $\mathcal{F}$ pseudo-shatters a set $S = (x_1, \ldots, x_{|S|})$ with witnesses $(r_1, \ldots, r_{|S|}) \in \mathbb{R}^{|S|}$ if $|(\mathrm{sign}(f(x_1) - r_1), \ldots, \mathrm{sign}(f(x_{|S|}) - r_{|S|})) : f \in \mathcal{F}| \ = \ 2^{|S|}$. Then, we define the pseudo-dimension of $(X, \mathcal{F})$, denoted $\mathrm{Pdim}(X, \mathcal{F})$, as the size of the largest set $S$ that can be pseudo-shattered by $\mathcal{F}$. If sets of arbitrary size can be pseudo-shattered, we say $\mathrm{Pdim}(X, \mathcal{F}) = +\infty$.*

Analogous to the VC dimension for binary classification, the pseudo-dimension provides upper bounds on the number of samples required to learn a function from the class $\mathcal{F}$ with high probability and low error. Specifically, for a given error $\epsilon$ and confidence $\delta$, the sample complexity of for $\mathcal{F}$ is $\widetilde{O}\big(\epsilon^{-2}(\mathrm{Pdim}(X, \mathcal{F}) + \log\frac{1}{\delta})\big)$ (Mohri et al., 2018, Theorem 19.2).

## A.3  COMPUTATIONAL COMPLEXITY

As discussed in Section 5.1, the algorithm TREE-COLSI is designed, rather than to minimize the computational complexity, to align with message passing and to allow to detect quite-colorful maps.

The algorithm, for a fixed node $u$ and level $\ell$, considers $|T^\ell|$ pattern graph's nodes. For each of such nodes, it tries $O(\Delta^{\Delta_T})$ sequences of nodes on line 13, where $\Delta$ is an upper bound to the degree of any node in $G$ and $\Delta_T$ is an upper bound to the number of children of any node in the pattern tree. A set $\mathcal{C}_{v,q}$ can contain in the worst case $Q = \sum_{i=0}^{|V_T|} \binom{|\Omega|}{i}$ colorsets on $|\Omega|$ colors. Then, for each sequence of nodes $(v_1, \ldots, v_\delta)$, the algorithm tries $O(Q^{\Delta_T})$ sequences of colorsets on line 14. Therefore, summing over all $u \in V_G$ and $\ell \in 0, \ldots, h$, we obtain a total of $O\left(\sum_{\ell=0}^h |T^\ell||V_G|\big(\Delta Q\big)^{\Delta_T}\right)$ $= O\left(|V_G||V_T| \cdot \big(\Delta Q\big)^{\Delta_T}\right)$ calls to MERGE.

In contrast, the original color coding algorithm (Alon et al., 2008), would have a complexity of $O\left(|E_G||V_T| \cdot Q\right) = O\left(|V_G||V_T| \cdot \Delta Q\right)$.

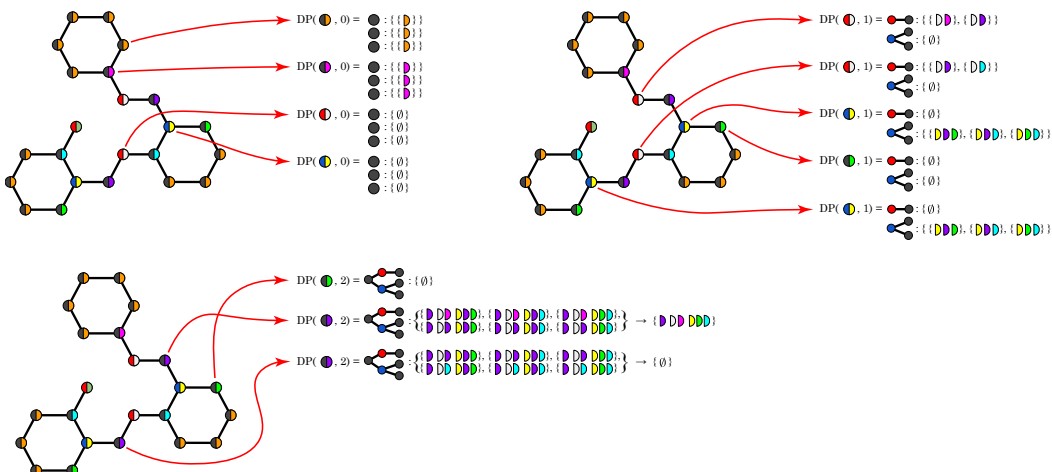

Figure 5: A simplified example of the execution of TREE-COLSI for a pattern $P$ and graph $G$.

A.4 AN EXAMPLE OF THE EXECUTION OF TREE-COLSI

## A.4 AN EXAMPLE OF THE EXECUTION OF TREE-COLSI

Figure 5 reports a (simplified) example of the execution of TREE-COLSI for a pattern $P$ and target graph $G$. The nodes of $G$ are endowed with labels (depicted on the left half of the node) and colors (depicted on the right half of the node) obtained by one iteration of color refinement. The original graph can be seen in Figure 3.

At level 0, the algorithm returns for each node $u \in V_G$ a dictionary from the three leaves of $P$ to sets of colorsets. We report as an example four such $u$'s. For nodes $u$ with black label, the algorithm keeps track of the colors of the images of the pattern graph's nodes (in fact, in the algorithm, these colors are maintained implicitly). For nodes $u$ with other labels, the algorithm returns an empty set since the sub-patterns don't match.

At level 1, the algorithm returns dictionaries from the two sub-patterns of height 1 of $P$ to sets of colorsets. For example, consider the first node $u_1$ we report (i.e., with red label and white color). Here, the sub-pattern rooted at the red-labeled node matches the red label of $u_1$, and the colorsets received from the two neighbors of $u_1$ both allow for the construction of a subgraph isomorphism map. Then, the algorithm keeps track of the sets of colors of the images of the pattern graph's nodes for the two maps. In particular, it keeps track of a set containing a white and pink color, and of a set containing a white and purple color.

Finally, at level 2, the algorithm returns dictionaries from the entire pattern $P$ to sets of colorsets. The first node $u_1$ we report (black label and green color), receives from its black-labeled neighbor empty sets. Therefore, it also returns an empty set. The second node $u_2$ we report (black label and purple color) receives from both its two neighbors some colorsets. Here, one such combination of colorsets is valid, which is then returned as a valid colorset. Since 2 is the height of the tree, it means that there exists a subgraph isomorphism from the tree pattern to $G$ rooted at $u_2$. The third node $u_3$ we report (black label and purple color) also receives from both its two neighbors some colorsets. However, in all such combinations of colorsets, some colors collide. Therefore, it returns an empty set. Indeed, even though there is a subgraph isomorphism rooted at $u_3$, it is not quite-colorful.

## B  EXTENSIONS TO THE DYNAMIC PROGRAMMING ALGORITHM

### B.1  A DYNAMIC PROGRAM FOR COLORFUL SUBTREE ISOMORPHISM

In Section 5.1, we obtained a message-passing-like dynamic program for *quite-colorful* subgraph isomorphism from tree patterns. However, the quite-colorful condition can be too restrictive in some scenarios. In this section, we tackle the locally injective homomorphism problem from tree patterns. In this relaxation of the subgraph isomorphism problem, the map from the pattern to the

target is required to be injective only on the neighborhood of each pattern graph's node. We then propose TREE-COLLIH, a slightly modified algorithm that solves the locally injective homomorphism problem under a weaker condition with respect to quite-colorfulness.

Inspired by the color coding technique, as done in Section 5.1, we enforce *local* injectivity by making sure that the colors of the images of specific pairs of pattern nodes are distinct. In particular, we now

---

**Algorithm 3:** PARENTMERGE$((C_1, \ldots, C_\delta))$

1 **if** $\exists i : c(u) \in C_i$: **return** $\epsilon$
2 **return** $\{c(v_i), i = 1, \ldots, \delta\}$

---

require only that for each node $p$, the image of the parent of its parent has a different color from the image of $p$. This condition is enforced by substituting the MERGE procedure of the dynamic program TREE-COLSI, reported in Algorithm 1 with PARENTMERGE. In particular, the dynamic program still returns, for nodes $u \in V_G$, $p \in V_T$, a set $\mathcal{C}_{u,p}$ of colorsets $C \in \{0, 1\}^{|\Omega|}$. If there is no locally injective homomorphism from $T_p$ to $G$ mapping $p$ to $u$ the set is empty. If $C \in \mathcal{C}_{u,p}$, then there exists (at least) one associated map $\phi$ from $T_p$ to $G$.

We now formally characterize the family of maps that can be recognized by the dynamic program.

**Definition 5.** *Let $T_r$ be a tree, $G$ be a graph whose nodes are endowed with colors $c : V_G \to \Omega$, and $\phi : V_{T_r} \to V_G$ a map. We say $\phi$ is parent-colorful (w.r.t. c) if $\forall p \in V_{T_r}$, $\forall q \in \text{children}(p)$ and $\forall t \in \text{children}(q)$ it holds that $c(\phi(p)) \neq c(\phi(t))$.*

**Theorem 6.** *Let $T_r$ be a tree of height $h$, $G$ be a graph whose nodes are endowed with colors $c : V_G \to \Omega$. Then, TREE-COLLIH$_c(u, h)[r] \neq \emptyset$ if and only if there is a parent-colorful locally injective homomorphism $\phi$ from $T_r$ to $G$ such that $\phi(r) = u$.*

A simple sufficient (but not necessary) condition to ensure that the algorithm solves the locally injective homomorphism problem is that the parent of the parent of any pattern node must have a different label from the node itself, as the map will be parent-colorful. Moreover, if the discovered homomorphism is (globally) injective, the dynamic problem solves the subgraph isomorphism problem. A sufficient condition for this is that the cycles in the target graph are long enough.

**Corollary 2.** *Let $T_r$ be a tree of height $h$, $G$ be a graph whose nodes are endowed with colors $c : V_G \to \Omega$. Let $T$ be such that $\forall p \in V_T$, $\forall q \in \text{children}(p)$ and $\forall t \in \text{children}(q)$ it holds that $L(p) \neq L(t)$. Let also $G$ be such that the minimum cycle length is at least $2h + 1$. Then, TREE-COLLIH$_c(u, h)[r] \neq \emptyset$ if and only if there is a subgraph isomorphism $\phi$ from $T$ to $G$ such that $\phi(r) = u$.*

In particular, if we restrict ourselves to tree patterns with no node such that the parent of its parent has its same label, the dynamic program solves the subgraph isomorphism problem if the target graphs are trees. Moreover, in many molecular graphs from organic chemistry, the minimum cycle length is 5. Therefore, on these graphs the dynamic program correctly solves the subgraph isomorphism problem for (parent-colorful) tree patterns of height at most 2.

## B.2 DEALING WITH CYCLIC PATTERNS

The algorithms described in the previous sections, as the original algorithms (Alon et al., 1995), can only deal with tree patterns. This is necessary due to the fact that the maps built on different subtrees can be merged recursively without conflicts.

One can deal with cyclic patterns, although with weaker guarantees, by using as a pattern the truncated universal cover of the pattern. Let $P$ be the pattern at hand and $u \in V_P$, then one would use $P' = U_u^l(P)$ as the new pattern, with $l \geq \max_v d_P(u, v)$. Then, using TREE-COLLIH one would obtain a positive answer if $P$ is a subgraph of $G$, as indeed there exists a locally injective homomorphism from $U_u^l(P)$ to $G$ for each $l$. However, there could be false positives. For TREE-COLSI, one would need to enrich the colorsets to ensure that tree nodes $p, q \in V_{U_u^l(P)}$ that correspond to the same node in $P$ are enforced to have images with the same color, and that tree nodes that correspond to different nodes in $P$ are enforced to have different colors.

## B.3 FROM SUBGRAPH DETECTION TO COUNTING

As argued in a follow-up to the original color-coding algorithm Alon et al. (2008), modifying the dynamic program to count maps rather than solve the decision problem is relatively straightforward,

and can be applied to all the presented versions of the algorithm. Indeed, we endow colorsets with a counter cnt that keeps track of the number of maps associated with the colorset. Then, when building the set $\mathcal{C}_{u,p}$ consider a choice of vertices $v_1, \ldots, v_\delta$ and a sequence of colorsets $C_1, \ldots, C_\delta$ with associated counters $\mathrm{cnt}_i, \ldots, \mathrm{cnt}_\delta$, which each representing the number of maps from $T_{q_i}$ to $G$ mapping $q_i$ to $v_i$. Let then the resulting merged colorset be $C \neq \epsilon$. The number of maps from $T_p$ to $G$ mapping $p$ to $u$ is then $\overline{\mathrm{cnt}} = \prod_i \mathrm{cnt}_i$. If $C$ already belongs to $\mathcal{C}_{u,p}$ with associated counter $\mathrm{cnt}_C$, we update the counter as $\mathrm{cnt}_C = \mathrm{cnt}_C + \overline{\mathrm{cnt}}$. Otherwise, we insert $C$ in $\mathcal{C}_{u,p}$ with $\mathrm{cnt}_C = \overline{\mathrm{cnt}}$. In this variant of the algorithm there would be an increase in the size of the MLP for a GNN, to store the counters, to be able to simulate the algorithm.

## C  PROOFS

### C.1  SECTION 3

**Proposition 1.** *Let $\mathcal{G}$ be a set of graphs such that $\forall G_1, G_2 \in \mathcal{G}$, $G_1 \not\simeq_{\mathrm{WL}} G_2$ and $|V_G| \leq n$, $\forall G \in \mathcal{G}$. Let $f : \mathcal{G} \to \mathbb{R}$ be any function. Then, there exists a function class $\mathrm{GNN}_{\ell=n}$ realized by a GNN model such that $f \in \mathrm{GNN}_{\ell=n}$.*

*Proof.* Since $\forall G_1, G_2 \in \mathcal{G}$, $G_1 \not\simeq_{\mathrm{WL}} G_2$, by Morris et al. (2023a, Proposition 9) the GNN model can realize a one-hot encoding $h_G$ for each graph $G$. Then, by appending a linear layer to the model such that $W h_G = f(G)$, we can realize $f$. $\qquad\square$

**Lemma 1.** *The function class $\mathrm{GNN}_\ell$ of Prop. 1 has pseudo-dimension $\mathrm{Pdim}(\mathcal{G}, \mathrm{GNN}_\ell) = |\mathcal{G}|$.*

*Proof.* We have $\mathrm{Pdim}(\mathcal{G}, \mathrm{GNN}_\ell) \leq |\mathcal{G}|$ by definition. Moreover, by Morris et al. (2023a, Proposition 9) and the fact that no two graphs are WL-isomorphic, the GNN model can realize any binary function on $\mathcal{G}$. Then, taking as witnesses a vector of $-1/2$, we obtain the lower bound. $\qquad\square$

### C.2  SECTION 4

**Theorem 1.** *There exists a set of graphs $\mathcal{G}$ such that $\forall G_1, G_2 \in \mathcal{G}$, $G_1 \not\simeq_{\mathrm{WL}} G_2$ and a function $f : \mathcal{V}_{\mathcal{G}} \to \mathbb{R}$ for which there exists no function class $\mathrm{GNN}_\ell^{\mathrm{node}}$ realized by a GNN model such that $f \in \mathrm{GNN}_\ell^{\mathrm{node}}$, $\forall \ell$.*

*Proof.* The theorem is proven by the two graphs $G_2, G_3$ in Figure 1, and $f(G, u)$ the rooted subgraph isomorphism function, with as pattern the triangle $K_3$ rooted in any of its nodes. Clearly $f(G_2, u) = 1$ and $f(G_3, u) = 0$. These two graphs have isomorphic universal covers. In particular, rooting the universal covers at $u \in V_{G_2}$ and $u \in V_{G_3}$ yields $U_u(G_2) \simeq U_u(G_3)$. We then have that $\mathrm{WL}^\ell(G_2, u) \simeq \mathrm{WL}^\ell(G_3, u)$, by (Krebs & Verbitsky, 2015). Then, the GNN cannot assign different outputs to $(G_2, u)$ and $(G_3, u)$ (Morris et al., 2019, Theorem 1). $\qquad\square$

**Lemma 2.** *Subgraph counting and induced subgraph counting are node-decomposable into rooted subgraph counting and rooted induced subgraph counting, respectively.*

*Proof.* Let $P$ be a graph and $p \in V_P$ any node. The (induced) subgraph counting function $f(G)$ maps a graph $G$ to $|\Phi_G|$, where $\Phi_G$ is the set of (induced) subgraph isomorphisms $\phi$ from $P$ to $G$. We let $g(G, u)$ be the rooted subgraph counting function (resp. the rooted induced subgraph counting function), that is $g(G, u) = |\Phi_{G,u}|$ with $\Phi_{G,u}$ the set of (induced) subgraph isomorphisms $\phi$ from $P$ to $G$ such that $\phi(p) = u$. We show that $f(G) = \sum_{u \in V_G} g(G, u)$.

We have that any $\phi \in \Phi_{G,u}$ is a (induced) subgraph isomorphism, so $\phi \in \Phi_G$. Conversely, if $\phi \in \Phi_G$, then $\phi \in \Phi_{G,\phi(p)}$ by definition. Therefore $\Phi_G = \bigcup_{u \in V_G} \Phi_{G,u}$. It then suffices to show that the sets $\Phi_{G,u}$ are disjoint. Indeed, let $\phi \in \Phi_{G,u}$, then $\phi(p) = u$. Therefore, $\phi \notin \Phi_{G,v}$ for any $v \neq u$, since $\phi(p) \neq v$. $\qquad\square$

Note that an alternative and equally valid definition of the (induced) subgraph counting function counts the number of (induced) subgraph isomorphisms up to automorphisms of the pattern graph. In this case, the function $f'(G)$ maps a graph to $|\{V \subseteq V_G : \text{exists (induced) subgraph isomorphism}$

$\phi$ from $P$ to $G$ such that $\phi(V_P) = V\}|$. Note that $f'(G) = f(G)/\text{Aut}(P)$. Also this function is node-decomposable, as $f'(G) = \sum_{u \in V_G} g'(G, u)$ with $g'(G, u) = g(G, u)/\text{Aut}(P)$.

**Theorem 2.** *Let $\mathcal{G}$ be a $(\ell, k)$-identifiable set of graphs. Consider any $k$-local function $f : \mathcal{V}_\mathcal{G} \to \mathbb{R}$. Then, there exists a function class $\text{GNN}_\ell^{\text{node}}$ realized by a GNN model with $O(\eta_{\ell,\mathcal{G}}^2 \cdot \ell)$ parameters and $\ell$ layers such that $f \in \text{GNN}_\ell^{\text{node}}$.*

*Proof.* Let $h_v^0$ be a one-hot encoding of node labels. Then, we apply (Morris et al., 2019, Theorem 2, Lemma 11) to a graph $G$ composed by the disjoint union of the graphs in $\mathcal{G}$. In particular, the GNN layers require the dimensionality of the node embeddings to be $\eta_{\ell,\mathcal{G}}$. Therefore, after $\ell$ layers with $O(\eta_{\ell,\mathcal{G}}^2)$ parameters each, we have that the embeddings $\hat{h}_u^\ell$ outputted by the $\ell$-th layer are such that $\hat{h}_u^\ell = \hat{h}_v^\ell$ iff $U_v^\ell(G) = U_u^\ell(G)$. Moreover, $\hat{h}_u^\ell \in \{-1, 1\}^{\eta_{l,\mathcal{G}}}$. Let $U_1, \ldots, U_{\eta_{l,\mathcal{G}}}$ be an arbitrary enumeration of the truncated universal covers in $\mathcal{G}$, and $h_1, \ldots, h_{\eta_{\ell,\mathcal{G}}}$ the corresponding embeddings.

Let $W \in \mathbb{R}^{\eta_{\ell,\mathcal{G}} \times \eta_{\ell,\mathcal{G}}}$ with $W_{i,:} = h_i^\top$. Let $b \in \mathbb{R}^{\eta_{\ell,\mathcal{G}}}$ with $b_i = -\eta_{l,\mathcal{G}} + 1$. We let $h_u^\ell = \text{ReLU}(W \hat{h}_u^\ell + b)$, then $h_u^\ell \in \mathbb{R}^{\eta_{\ell,\mathcal{G}}}$ is a one-hot encoding for $U_u^\ell(G)$. Therefore, by appending a linear layer we can assign any real value to $U_u^\ell(G)$.

If, for any two $(G_1, u), (G_2, v) \in \mathcal{V}_\mathcal{G}$, it holds that $\text{EGO}_u^k(G_1) \not\simeq \text{EGO}_v^k(G_2)$, then we have that $U_u^\ell(G_1) \not\simeq U_u^\ell(G_2)$ by $(\ell, k)$-identifiability, and therefore the GNN model can assign different values to these two ego-nets. In particular, since $f$ is $k$-local, we can realize $f$.

Finally, the number of parameters of the model is $O(\eta_{\ell,\mathcal{G}}^2 \cdot \ell)$. $\qquad\square$

**Corollary 1.** *Let $\mathcal{G}$ be a $(\ell, k)$-identifiable set of graphs. Let $\text{GNN}_\ell$ be a function class realized by a GNN model with sum-aggregation $f_{\text{out}}(\{\!\{ h_u^\ell : u \in V_G \}\!\}) = \sum_{u \in V_G} h_u^\ell$. Then $\text{GNN}_\ell$ can perform both subgraph counting and induced subgraph counting of patterns of radius at most $k$.*

*Proof.* Both the rooted subgraph counting and rooted induced subgraph counting of patterns of radius at most $k$ are $k$-local functions, and we can apply Theorem 2. Then, the sum aggregation computes the subgraph counting and induced subgraph counting functions at the graph level. $\quad\square$

**Theorem 3.** *The function class $\text{GNN}_\ell$ of Cor. 1 has pseudo-dimension $\text{Pdim}(\text{GNN}_\ell) \leq \eta_{\ell,\mathcal{G}} + 1$.*

*Proof.* Consider a function $f \in \text{GNN}_\ell$, and the associated function $g \in \text{GNN}_\ell^{\text{node}}$ that produces the node-level opuputs that are then aggregated via $f_{\text{out}}$ to produce $f$. We show $f(G)$ can be written as $f(G) = w^\top x_G$, with $x_G \in \mathbb{R}^{\eta_{\ell,\mathcal{G}}}$ the multiplicity vector of the truncated universal covers of height $\ell$ rooted at the nodes of $G$. In particular, let $U_1, \ldots, U_{\eta_{\ell,\mathcal{G}}}$ an enumeration of the truncated universal covers. Then, the $i$-th entry of $x_G$ will be $k$ iff there are $k$ nodes $u \in V_G$ such that $U_u^\ell(G) \simeq U_i$. Note that $x_{G_1} = x_{G_2}$ if and only if $G_1 \simeq_{\text{WL}_\ell} G_2$. Therefore, we can set the $i$-th entry of the vector $w$ to the output of the model $g(G, u) \in \text{GNN}_\ell^{\text{node}}$ on a node $u$ that has rooted universal cover $U_i$. Then, $f(G) = \sum_{u \in V_G} g(G, u) = w^\top x_G$.

By Anthony & Bartlett (1999, Theorem 11.6), the pseudo-dimension of a linear function on $\mathbb{R}^d$ is $d + 1$. Then, by Anthony & Bartlett (1999, Corollary 11.5), we have that $\text{Pdim}(\text{GNN}_\ell) \leq \eta_{\ell,\mathcal{G}} + 1$. $\qquad\square$

## C.3 SECTION 5.1

**Lemma 3.** *Let $G$ be a graph, $T_r$ be a tree of height $h$. Let nodes $V_G$ be endowed with colors $c$. Let $\phi$ be a quite-colorful subgraph isomorphism from $T$ to $G$. Then $\mathcal{C}_{\phi(r),r} \neq \emptyset$.*

*Proof.* Given a node $p \in V_T$, we call $T_p$ the subtree of $T_r$ rooted at $p$. We show inductively that the colorset $\{c(\phi(q)) : q \in V_{T_p} \setminus \{p\}\}$ belongs to $\mathcal{C}_{\phi(p),p}$.

We first address the base case of the dynamic program, $p$ is a leaf. For each leaf $p \in V_T$, we have that $\mathcal{C}_{\phi(p),p} = \{\emptyset\}$, since $L(\phi(p)) = L(p)$.

Let now $p$ be a non-leaf. Then, there exists a sequence $v_1, \ldots, v_\delta$ of distinct neighbors of $\phi(p)$ such that $v_i = \phi(q_i), q_i \in \text{children}(p)$ and, for each $q$, $\phi|_{V_{T_q}}$ is a quite colorful subgraph isomorphism from $T_q$ to $G$. We then have inductively that $C_q := \{c(\phi(w)) : w \in V_{T_q} \setminus \{q\}\} \in \mathcal{C}_{\phi(q),q}$, $\forall q \in \text{children}(p)$.

Since $L(\phi(p)) = L(p)$, the algorithm starts creating the set $\mathcal{C}_{\phi(p),p}$ in line 12. In particular, it will try on line 13 the correct sequence of nodes $(v_1, \ldots, v_\delta) = (\phi(q_1), \ldots, \phi(q_\delta))$ and for that sequence the algorithm will try on line 16 the sequence of colorsets $C_{q_1}, \ldots, C_{q_\delta}$. For any two distinct $q_1, q_2$, and $\forall t_1 \in V_{T_{q_1}} \setminus \{q_1\}, t_2 \in V_{T_{q_2}} \setminus \{q_2\}$, we have that $d_T(t_1, t_2) > 3$. Since $\phi$ is quite-colorful, the third flag condition is true. Moreover, $d_T(q_1, t_2) \geq 3$, and the second flag condition is true. Finally, either $d_T(p, t_2) \geq 3$ or $p = \text{parent}(\text{parent}(t_2))$, and the first flag condition is true. Therefore, all three conditions for flag are true and the algorithm inserts in $\mathcal{C}_{\phi(p),p}$ the colorset $C := \bigcup_{q \in \text{children}(p)} (C_q \cup c(\phi(q))) = \{c(\phi(q)) : q \in V_{T_p} \setminus \{p\}\}$.

Then, we have inductively that $C_r \in \mathcal{C}_{\phi(r),r}$, and we have the claim. $\square$

**Lemma 4.** *Let $G$ be a graph, $T_r$ be a tree of height $h$. Let nodes $V_G$ be endowed with colors $c$. If $\mathcal{C}_{u,r} \neq \emptyset$, then there exists a quite-colorful subgraph isomorphism $\phi$ from $T$ to $G$ such that $\phi(r) = u$.*

*Proof.* Given a node $p \in V_T$, we call $T_p$ the subtree of $T_r$ rooted at $p$. We show inductively that if $C \in \mathcal{C}_{u,p}$, then there exists a quite-colorful subgraph isomorphism $\phi$ from $T_p$ to $G$ such that $\phi(p) = u$ and $\{c(\phi(q)) : q \in V_{T_p} \setminus \{p\}\} = C$.

We first address the base case of the dynamic program, $p$ is a leaf. For each leaf $p \in V_T$, we have that $\mathcal{C}_{u,p} = \{\emptyset\} \neq \emptyset$ iff $L(u) = L(p)$, that is if $p \mapsto u$ is a (quite-colorful) subgraph isomorphism.

Let now $C \in \mathcal{C}_{u,p}$ for some non-leaf $p$. Then, there exists a sequence of distinct nodes $(v_1, \ldots, v_\delta)$ from $\mathcal{N}(u)$ such that $C$ was obtained by a sequence of color sets $(C_1, \ldots, C_\delta)$, with $C_i \in \mathcal{C}_{v_i,q_i}, \forall i = 1, \ldots, \delta$, such that all three conditions for flag are true.

Then, we have inductively that there exists an isomorphism $\phi_i$ from $T_{q_i}$ to $G$ such that $\phi_i : q_i \mapsto v_i$ and $\{c(\phi_i(t) : t \in V_{T_{q_i}} \setminus \{q_i\})\} = C_i$, for each $q_i \in \text{children}(p)$. The domains of such isomorphisms are all distinct, since the sets $V_{T_q} : q \in \text{children}(p)$ are pairwise disjoint. We can therefore define a new map $\phi : V_{T_p} \to V_G$ as $\phi|_{V_{T_{q_i}}} = \phi_i$ and $\phi : p \mapsto u$. This map is a homomorphism. Indeed, each $\phi_i$ is an homomorphism and therefore maps adjacent nodes in $V_{T_{q_i}}$ to adjacent nodes in $V_G$. Moreover, each $q_i \in \text{children}(p)$ is mapped to a node $v_i \in \mathcal{N}(u)$.

To show that $\phi$ is a subgraph isomorphism, we just need to show injectivity by showing that the co-domains $H_i = \{\phi(w) : w \in V_{T_{q_i}}\}$ of the $\phi_i$ and the set $\{u\}$ are all pairwise disjoint. We show first that $H_i \cap H_j = \emptyset$. In particular, by flag condition (3) we have that $C_i \cap C_j = \emptyset$, and therefore $z \notin \{\phi(w) : w \in V_{T_{q_j}} \setminus \{q_j\}\}$ for each $z \in \{\phi(w) : w \in V_{T_{q_i}} \setminus \{q_i\}\}$. Moreover, $c(v_i) \notin C_j$ by flag condition (2), and therefore $\phi(q_i) = v_i \notin \{\phi(w) : w \in V_{T_{q_j}} \setminus \{q_j\}\}$. Also by flag condition (2), $c(v_j) \notin C_i$, so $\phi(q_j) = v_j \notin \{\phi(w) : w \in V_{T_{q_i}} \setminus \{q_i\}\}$. Finally, $\phi(q_i) = v_i \neq v_j = \phi(q_j)$. Thus, $H_i \cap H_j = \emptyset$. We now show that $u \notin H_i, \forall i$. First, we have by flag condition (1) that $c(u) \notin C_i, \forall i$ and therefore $u \notin \{\phi(w) : w \in V_{T_{q_i}} \setminus \{q_i\}\}$. Moreover, $u \neq v_i = \phi(q_i)$. Therefore, $\phi$ is injective and is a subgraph isomorphism.

Moreover, we have $\{c(\phi(q)) : q \in V_{T_p} \setminus \{p\}\} = \bigcup_q \{c(\phi_i(t) : t \in V_{T_q} \setminus \{q\})\} \cup \{c(\phi(q))\} = C$. Finally, since the colors of the images of the nodes in $T_p$ respect the flag conditions, $\phi$ is quite-colorful.

Then, we have inductively that if $C \in \mathcal{C}_{u,r}$, there exist a quite-colorful subgraph isomorphism $\phi$ from $T$ to $G$ such that $\phi(r) = u$, and we have the claim. $\square$

**Theorem 4.** *Let $T_r$ be a tree of height $h$, $G$ be a graph whose nodes are endowed with colors $c : V_G \to \Omega$. Then, TREE-COLSI$_c(u, h)[r] \neq \emptyset$ if and only if there is a quite-colorful subgraph isomorphism $\phi$ from $T_r$ to $G$ such that $\phi(r) = u$.*

*Proof.* If there is a quite-colorful subgraph isomorphism $\phi$ from $T_r$ to $G$ such that $\phi(r) = u$, then by Lemma 3 we have that $\mathcal{C}_{\phi(r),r} \neq \emptyset$. Moreover, by Lemma 4, if $\mathcal{C}_{u,r} \neq \emptyset$, then there exists a quite-colorful subgraph isomorphism $\phi$ from $T_r$ to $G$ such that $\phi(r) = u$. $\qquad\square$

## C.4  SECTION 5.2

**Theorem 5.** *Let $\mathcal{G}$ be a set of graphs of bounded degree, $T_r$ be a tree of height $h$. Let $f(G) = 1$ if $\exists u \in V_G : \textsc{Tree-colSI}_c(u,h)[r] \neq \emptyset$ and 0 otherwise. Then, there exists a function class $\mathrm{GNN}_{l+h}$ realized by a GNN model with $l + h$ layers and $O\left(\eta_{l,\mathcal{G}}^2 \cdot l + \zeta_{l,T_r,\mathcal{G}} \cdot h\right)$ parameters such that $f \in \mathrm{GNN}_{l+h}$.*

*Proof.* We simulate the DP algorithm with a GNN. We first use $l$ layers to obtain colors. Then, we use $h$ layers to simulate the $h$ recursive calls of the DP. Let the graphs be such that $\forall u \in V_G, |\mathcal{N}(u)| < \Delta$.

**Color construction**   We consider the GNNs of Section A.1. Let $h_v^0$ be a one-hot encoding of node labels. Then, we use (Morris et al., 2019, Theorem 2, Lemma 11) applied to a graph $G$ composed by the disjoint union of the graphs in $\mathcal{G}$. In particular, the GNN layers require the dimensionality of the embeddings to be $\eta_{l,\mathcal{G}}$. Therefore, after $l$ layers with $O(\eta_{l,\mathcal{G}}^2)$ parameters each, we have that the embeddings $\bar{h}_u^l$ outputted by the $l$-th layer are such that $\bar{h}_u^l = \bar{h}_v^l$ iff $U_v^l(G) = U_u^l(G)$. Moreover, $\bar{h}_u^l \in \{-1, 1\}^{\eta_{l,\mathcal{G}}}$. We take these embeddings as the colors $c$ used by the dynamic program, and let $\bar{h}_1, \ldots, \bar{h}_{\eta_{l,\mathcal{G}}}$ be an enumeration of them.

Let $\mathcal{C}_1, \ldots, \mathcal{C}_D$ be an enumeration of the possible elements $(\mathcal{C}_{u,\ell}, c(u))$, i.e., for each possible $u$ and $\ell$. We suppose, without loss of generality, that if node $u$ has color $c(u) = \bar{h}_i$, then $(\mathcal{C}_{u,\ell=0}, c(u)) = \mathcal{C}_i$, for each $i = 1, \ldots, \eta_{l,\mathcal{G}}$. This is valid since $\mathcal{C}_{u,\ell=0}$ is uniquely determined by $c(u)$.

Let $W^{(1)} \in \mathbb{R}^{\eta_{l,\mathcal{G}} \times \eta_{l,\mathcal{G}}}$ with $W_{i,:} = \bar{h}_i^\top$. Let $b \in \mathbb{R}^{\eta_{l,\mathcal{G}}}$ with $b_i = -\eta_{l,\mathcal{G}} + 1$. Then $\mathrm{ReLU}(W^{(1)}\bar{h}_u^l + b)$ is the one-hot encoding for the color of $u$. We take $W^{(2)} \in \mathbb{R}^{1 \times \eta_{l,\mathcal{G}}}$, with $W_{1,i}^{(2)} = \Delta^{i-1}$, and let $h_u^l = W^{(2)}\mathrm{ReLU}(W^{(1)}\bar{h}_u^l + b) \in \mathbb{R}$. This is obtained by appending a two-layer MLP to the MLP that outputs $\bar{h}_u^l$. We then have that $h_u^l = \Delta^{i-1}$ if and only if $(\mathcal{C}_{u,\ell=0}, c(u)) = \mathcal{C}_i$.

**Message aggregation**   Then, the following $h$ GNN layers have to simulate the dynamic program. Suppose that $h_v^{l+\ell-1} = \Delta^{i-1}$ if and only if $(\mathcal{C}_{v,\ell-1}, c(v)) = \mathcal{C}_i$. This is true by construction for $\ell = 1$. Then we have that $\hat{h}_u^{l+\ell} = \left(h_u^{l+\ell-1}, \sum_{v \in \mathcal{N}(u)} h_v^{l+\ell-1}\right) \in \mathbb{R}^2$ is a pair of integers, obtained via the sum-aggregation of the messages from $\mathcal{N}(u)$, that is a unique identifier for the element $\mathcal{E} = ((\mathcal{C}_{u,\ell-1}, c(u)), \{\!\!\{(\mathcal{C}_{v,\ell-1}, c(v)) : v \in \mathcal{N}(u)\}\!\!\})$. Let $\mathcal{E}_1, \ldots, \mathcal{E}_{\zeta_{l,T_r,\mathcal{G}}}$ be an enumeration of the possible elements $\mathcal{E}$, and let $\hat{h}_1, \ldots, \hat{h}_{\zeta_{l,T_r,\mathcal{G}}} \in \mathbb{R}^2$ be the associated vectors.

**Mapping aggregated messages to DP states**   Not that, in the dynamic program, the computation of a set $\mathcal{C}_{u,\ell}$ depends uniquely on the sets $\mathcal{C}_{v,\ell-1}$ for all $v \in \mathcal{N}(u)$, as well as the colors $c(v)$ for $v \in \mathcal{N}(u) \cup \{u\}$. Indeed, while also the labels $L(u)$ are used, these are uniquely identified by the colors $c(u)$ by construction. We then show that we can simulate the function mapping $((\mathcal{C}_{u,\ell-1}, c(u)), \{\!\!\{(\mathcal{C}_{v,\ell-1}, c(v)) : v \in \mathcal{N}(u)\}\!\!\})$ to $(\mathcal{C}_{u,\ell}, c(v))$ using a MLP.

Note that the map $\hat{h}_u^{l+\ell} \mapsto [\Delta^{\zeta_{l,T_r,\mathcal{G}}}, 1]\hat{h}_u^{l+\ell} \in \mathbb{R}$ is injective. Let then $W^{(1)} \in \mathbb{R}^{\zeta_{l,T_r,\mathcal{G}} \times 2}$ with $W_{i,:}^{(1)} = [\Delta^{\zeta_{l,T_r,\mathcal{G}}}, 1]$ and $b \in \mathbb{R}^{\zeta_{l,T_r,\mathcal{G}}}$ with $b_i = -[\Delta^{\zeta_{l,T_r,\mathcal{G}}}, 1]\,\hat{h}_i$. Then $\mathrm{tri}(W^{(1)}\hat{h}_u^{l+\ell} + b)$ is a one-hot encoding for $((\mathcal{C}_{u,\ell-1}, c(u)), \{\!\!\{(\mathcal{C}_{v,\ell-1}, c(v)) : v \in \mathcal{N}(u)\}\!\!\})$.

Let then $W^{(2)} \in \mathbb{R}^{1 \times \zeta_{l,T_r,\mathcal{G}}}$, with $W_{1,i}^{(2)} = \Delta^{j-1}$ such that $\mathcal{C}_j$ is the output of one iteration of the dynamic program (i.e., lines 6 to 21) when it receives as input $\mathcal{E}_i$. Then $h_u^{l+\ell} = W^{(2)}\mathrm{tri}(W^{(1)}\hat{h}_u^{l+\ell} + b)$ is equal to $\Delta^{i-1}$ if and only if $(\mathcal{C}_{u,\ell}, c(v)) = \mathcal{C}_i$. Then, by induction on $\ell$, we have that $h_u^{l+h}$ is a unique identifier for $(\mathcal{C}_{u,h}, c(u))$.

**Output layer** In fact, we modify the last layer MLP by choosing $W^{(2)}$ with $W^{(2)}_{1,i} = 1$ if the output of one iteration of the dynamic program when it receives as input $\mathcal{E}_i$ is an element $(\mathcal{C}_{u,h}, c(u))$ with $\mathcal{C}_{u,h}[r] \neq \emptyset$, and 0 otherwise. Then $h^{l+h}_u = 1$ if and only if $\mathcal{C}_{u,h}[r] \neq \emptyset$. Note that we obtained a valid simulation of the dynamic program by using $h$ GNN layers as defined in Section A.1, at the node level. Finally, if we take as $f_{\text{out}}(\{\!\{h^{l+h}_u : u \in V_G\}\!\}) = \text{lsig}(\sum_{u \in V_G} h^{l+h}_u)$, we have that the model realizes the function $f(G) = 1$ if $\exists u \in V_G : \text{TREE-COLSI}(u, h)[r] \neq \emptyset$ and 0 otherwise.

We observe that the MLPs use either an triangle function $\text{tri}$ or a linearized sigmoid $\text{lsig}$ function. Both can be simulated using ReLU as follows.

Let $x \in \mathbb{R}$. Then $\text{tri}(x) = W^{(2)}\text{ReLU}(W^{(1)}x + b)$ with $W^{(1)} = [1, 1, 1]^\top \in \mathbb{R}^{3 \times 1}$, $b = [1, 0, -1]^\top \in \mathbb{R}^3$ and $W^{(1)} = [1, -2, 1] \in \mathbb{R}^{1 \times 3}$.

Let $x \in \mathbb{R}$. Then $\text{lsig}(x) = W^{(2)}\text{ReLU}(W^{(1)}x + b)$ with $W^{(1)} = [1, 1]^\top \in \mathbb{R}^{2 \times 1}$, $b = [0, -1]^\top \in \mathbb{R}^2$ and $W^{(1)} = [1, -1] \in \mathbb{R}^{1 \times 2}$.

Moreover, in the proof, some parameters require a non-constant number of bits. We argue that such parameters can be replaced by more parameters with constant number of bits. Indeed, it is enough to maintain the invariant that $h^{l+\ell-1}_v \in \mathbb{R}^{\zeta_{l,T_r,\mathcal{G}}}$ is an one-hot encoding for $\mathcal{C}_i$ rather than $\Delta^{i-1}$. In this case, the number of parameters for each MLP grows from $O(\zeta_{l,T_r,\mathcal{G}})$ to $O(\zeta^2_{l,T_r,\mathcal{G}})$.

$\square$

**Lemma 5.** *Let $\mathcal{G}$ be a set of graphs such that $\forall u \in V_G$, $|\mathcal{N}(u)| < \Delta$. Let $T$ be a tree with $|V_T| = \kappa$. The number $\zeta_{l,T,\mathcal{G}}$ of distinct elements $\mathcal{E} = ((\mathcal{C}_{u,\ell-1}, c(u)), \{\!\{(\mathcal{C}_{v,\ell-1}, c(v)) : v \in \mathcal{N}(u)\}\!\})$ satisfies*

$$\zeta_{l,T,\mathcal{G}} \in O\left( \min\left( \eta_{l+h,\mathcal{G}} , \ \eta^{\Delta+1}_{\mathcal{G},l}/\Delta! \cdot 2^{(\Delta+1)\eta^\kappa_{\mathcal{G},l}/(\kappa-1)!} \right) \right).$$

*Proof.* There are $\kappa$ nodes in the pattern, and $\eta_{\mathcal{G},l}$ distinct colors. Therefore, there are $Q \leq \sum_{i=0}^{\kappa} \binom{\eta_{\mathcal{G},l}}{i} = O(\eta^\kappa_{\mathcal{G},l}/\kappa!)$ possible colorsets $C$.

Then, we have at most $2^Q$ sets of colorsets $\mathcal{C}_{u,p}$ and $2^{\kappa Q}$ sets of colorsets $\mathcal{C}_{u,\ell}$. In turn, there can be at most $D = \eta_{\mathcal{G},l} \cdot 2^{\kappa Q}$ elements $(\mathcal{C}_{u,\ell}, c(u))$. Therefore, we have $D = O(\eta_{\mathcal{G},l} \cdot 2^{\kappa \eta^\kappa_{\mathcal{G},l}/\kappa!})$. Then, there are at most $\zeta_{l,T,\mathcal{G}} = O(D^\Delta/\Delta! \cdot D) = O(\frac{\eta^{\Delta+1}_{\mathcal{G},l}}{\Delta!} \cdot 2^{(\Delta+1)\eta^\kappa_{\mathcal{G},l}/(\kappa-1)!})$ elements $\mathcal{E}$.

Moreover, the element $\mathcal{E} = ((\mathcal{C}_{u,\ell-1}, c(u)), \{\!\{(\mathcal{C}_{v,\ell-1}, c(v)) : v \in \mathcal{N}(u)\}\!\})$ is fully characterized by $U^{l+h}_u(G)$, for any $\ell = 1, \ldots, h$, so we also have $\zeta_{l,T,\mathcal{G}} \leq \eta_{l+h,\mathcal{G}}$.

$\square$

### C.5 SECTION B.1

**Lemma 6.** *Let $G$ be a graph, $T_r$ be a tree of height $h$. Let nodes $V_G$ be endowed with colors $c$. Let $\phi$ be a parent-colorful locally injective homomorphism from $T$ to $G$. Then $\mathcal{C}_{\phi(r),r} \neq \emptyset$.*

*Proof.* Given a node $p \in V_T$, we call $T_p$ the subtree of $T$ rooted at $p$. Let $\phi$ be a parent-colorful locally injective homomorphism from $T_p$ to $G$. We show inductively that $C \in \mathcal{C}_{\phi(p),p}$ with $C = \{c(\phi(q)) : q \in \text{children}(p)\}$.

We first address the base case of the dynamic program, $p$ is a leaf. For each leaf $p \in V_T$, we have that $\mathcal{C}_{\phi(p),p} = \{\emptyset\}$, since $L(\phi(p)) = L(p)$.

Let now $p$ be a non-leaf. Then, there exists a sequence $v_1, \ldots, v_\delta$ of distinct neighbors of $\phi(p)$ such that $\phi(q_i) = v_i, q_i \in \text{children}(p)$ and, for each $q$, $\phi|_{V_{T_q}}$ is still a parent-colorful locally injective homomorphism from $T_q$ to $G$. We have inductively that $C_q \in \mathcal{C}_{\phi(q),q}$, with $C_q$ the colorset associated with $\phi|_{V_{T_q}}$, i.e, $\{c(\phi(t)) : t \in \text{children}(q)\}$.

Since $L(\phi(p)) = L(p)$, the algorithm starts creating the set $\mathcal{C}_{\phi(p),p}$ in line 12. In particular, it will try on line 13 the correct sequence of nodes $(v_1, \ldots, v_\delta) = (\phi(q_1), \ldots, \phi(q_\delta))$ and for that sequence the algorithm will try on line 16 the sequence of colorsets $C_{q_1}, \ldots, C_{q_\delta}$. Since $\phi$ is parent-colorful,

for each $q$ and $t \in \text{children}(q)$ we have that $c(\phi(p)) \neq c(\phi(t))$. Then, inductively, $c(\phi(p)) \notin C_q$ for each $q$. Therefore, the set $C_p = \{c(q) : q \in \text{children}(p)\}$ is inserted in $\mathcal{C}_{\phi(p),p}$.

Then we have inductively that $C_r \in \mathcal{C}_{\phi(r),r}$, and we have the claim. $\qquad\square$

**Lemma 7.** *Let $G$ be a graph, $T_r$ be a tree of height $h$. Let nodes $V_G$ be endowed with colors $c$. If $\mathcal{C}_{u,r} \neq \emptyset$, then there exists a parent-colorful locally injective homomorphism $\phi$ from $T$ to $G$ such that $\phi(r) = u$.*

*Proof.* Given a node $p \in V_T$, we call $T_p$ the subtree of $T$ rooted at $p$. We show inductively that if $C \in \mathcal{C}_{u,p}$, then there exists a parent-colorful locally injective homomorphism $\phi$ from $T_p$ to $G$ such that $\phi(p) = u$ and $\{c(\phi(q)) : q \in \text{children}(p)\} = C$.

Let $p$ be a leaf. Then we have that $\mathcal{C}_{u,p} = \{\emptyset\}$ iff $L(u) = L(v)$, that is $p \mapsto u$ is a (parent-colorful) locally injective homomorphism.

Let now $C \in \mathcal{C}_{u,p}$ for some non-leaf $p$. Then, there exists a sequence of distinct nodes $(v_1, \ldots, v_\delta)$ from $\mathcal{N}(u)$ such that $C$ was obtained by a sequence of color sets $(C_1, \ldots, C_\delta)$, with $C_i \in \mathcal{C}_{v_i,q_i}$, such that $c(u) \notin C_i, \forall i$.

Then, we have inductively that, for each $q_i \in \text{children}(p)$, there exists a locally injective homomorphism $\phi_i$ from $T_{q_i}$ to $G$ such that $\phi_i(q_i) = v_i$ and there is no node $t \in \text{children}(q_i)$ with $c(\phi(t)) = c(u)$.

The domains of such maps are all distinct, since the sets $V_{T_q} : q \in \text{children}(p)$ are pairwise disjoint. We can therefore define a new map $\phi : V_{T_p} \to V_G$ as $\phi|_{V_{T_{q_i}}} = \phi_i$ and $\phi(p) = u$. This is a homomorphism as children of $p$ are mapped to neighbors of $u$.

We show that $\phi$ is locally injective for each node $q \in V_{T_p}$. Note that, for each $i$, for nodes in $V_{T_{q_i}} \setminus \{q_i\}$ the connectivity is the same as the one given by $\phi_i$, and the claim therefore is true inductively. Moreover, since the nodes $v_1, \ldots, v_\delta = \phi(q_1), \ldots, \phi(q_\delta)$ are all distinct, $\phi$ is locally injective on $p$.

Moreover, for $q_i, \forall i$, we have that $\mathcal{N}(q_i) = \text{children}(q_i) \cup \{p\}$. Clearly $\phi(t_1) \neq \phi(t_2)$, $\forall t_1, t_2 \in \text{children}(q_i)$ since $\phi$ is locally injective on $V_{T_{q_i}}$. Finally, $\phi(t) \neq \phi(p) = u$ $\forall t \in \text{children}(q_i)$ since there is no node $t \in \text{children}(q_i)$ with $c(\phi(t)) = c(u)$. Because of this, the map $\phi$ is parent-colorful. Moreover, we have that $\{c(\phi(q)) : q \in \text{children}(p)\} = C$.

$\qquad\square$

**Theorem 6.** *Let $G$ be a graph, $T_r$ a tree of height $h$. Let nodes $V_G$ be endowed with colors $c$. Then, $\text{TREE-COLLIH}_c(u,h)[r] = \mathcal{C}_{u,r} \neq \emptyset$ if and only if there is a parent-colorful locally injective homomorphism $\phi$ from $T_r$ to $G$ such that $\phi(r) = u$.*

*Proof.* If there is a parent-colorful locally injective homomorphism $\phi$ from $T_r$ to $G$ such that $\phi(r) = u$, then by Lemma 6 we have that $\mathcal{C}_{\phi(r),r} \neq \emptyset$. Moreover, by Lemma 7, if $\mathcal{C}_{u,r} \neq \emptyset$, then there exists a locally injective homomorphism $\phi$ from $T_r$ to $G$ such that $\phi(r) = u$. $\qquad\square$

**Corollary 2.** *Let $T_r$ a tree of height $h$, $G$ be a graph whose nodes are endowed with colors $c$. Let $T$ be such that $\forall p \in V_T$, $\forall q \in \text{children}(p)$ and $\forall t \in \text{children}(q)$ it holds that $L(p) \neq L(t)$. Let also $G$ be such that the minimum cycle length is at least $2h + 1$. Then, $\text{TREE-COLLIH}_c(u,h)[r] \neq \emptyset$ if and only if there is a subgraph isomorphism $\phi$ from $T$ to $G$ such that $\phi(r) = u$.*

*Proof.* Let the colors of nodes in $G$ be their labels. Then, any locally injective homomorphism $\phi$ from $T$ to $G$ such that $\phi(r) = u$ is parent-colorful since $\forall p \in V_T$, $\forall q \in \text{children}(p)$ and $\forall t \in \text{children}(q)$ it holds that $L(p) \neq L(t)$ and therefore $c(\phi(p)) \neq c(\phi(t))$.

We apply Theorem 6. We then obtain that $\mathcal{C}_{u,r} \neq \emptyset$ if and only if there is a locally injective homomorphism $\phi$ from $T$ to $G$ such that $\phi(r) = u$. We need to show injectivity. Suppose by contradiction that $\phi(q_1) = \phi(q_2)$ for some $q_1 \neq q_2 \in V_T$. Let $p$ be the lowest common ancestor of $q_1$ and $q_2$. Let $(p, t_1, \ldots, t_{h_1} = q_1)$ and $(p, s_1, \ldots, s_{h_2} = q_2)$ be the paths connecting $p$ to respectively $q_1$ and $q_2$. Since $h_1, h_2 \leq h$, we have $h_1 + h_2 \leq 2h$, for any $q_1, q_2$. Then, $(\phi(p), \phi(t_1), \ldots, \phi(t_{h_1}) =$

$\phi(s_{h_2}), \ldots, \phi(s_1), \phi(p))$ is a cycle of length at most $2h$. This is a contradiction as the minimum cycle length is at least $2h + 1$. □

# D  ADDITIONAL EXPERIMENTAL RESULTS

In this section, we provide further experimental results and supplement the experimental section in the main paper. Our code and data are available at github.com/BorgwardtLab/GNNsCanCountSubstructures.

## D.1  COUNTING IN REAL-WORLD DATASETS

In order to showcase the subgraph counting capabilities of GNNs, we selected a subset of molecular datasets. We focus on molecular benchmark datasets since subgraph mining and counting in molecular data have been the focus of extensive research, due to the relevance of subgraphs corresponding to functional groups. Such subgraphs play an essential role in generating molecular fingerprints.

Statistical properties of all considered datasets can be found in Table 5. We note that some datasets include node attribute vectors encoding subgraph information (e.g., the 6-cycle). To mitigate such biases, we retain only the atom type as node feature in the datasets ogbg-molhiv, ogbg-molpcba, Peptides-func, and PCQM-Contact, also ensuring consistency with other datasets. All experiments and statistics provided in this work are based on such reduced node features.

To identify a set of suitable pattern graphs, we used the subgraph miner FSG (Kuramochi & Karypis, 2004), to mine all patterns that occur with a frequency of at least 25% across all considered datasets. We include all such tree patterns of size 5 and 6 as well as cyclic patterns of size 6 and 7. The condition on the frequency was implemented to ensure that subgraph counts are non-zero in a substantial number of cases. To complement the set of patterns with an additional cyclic graph, we furthermore include the (non-frequent) 5-cycle, which is often of particular interest in molecular structures. Since node attributes vary between the considered datasets, in Table 4, we depict a pattern graph $H$ with node colors such that in each dataset the corresponding pattern is obtained by replacing the colors of $H$ with specific node attributes.

We use the $\text{GNN}_K$ architecture as described in Section A.1, with $K = 4$ MLP-layer-based message passing layers. As some datasets include edge attributes, we modified the model to aggregate and append edge attributes to the node features during the message passing process. The dimensionality of the GNN embeddings is fixed at 512. We used the Adam optimizer with a variable learning rate and a batch size of 128. The data is split into 80% for training and 20% for testing. Finally, we train for 300 epochs and report the mean as well as the standard deviations over at total of 5 such runs.

We framed the learning problem as a classification task, where the classes correspond to discrete count values ranging from zero to the maximum count observed in the training set. As evaluation

Table 4: Test set results for subgraph counting with a GNN on molecular graphs. Node colors visualize different atom types. Reported: Normalized Mean Absolute Error (nMAE, see Def. in Sect. D.1) and Area Under the Curve (AUC) for the multi-class classification problem.

| Dataset | Metric | P1 | P2 | P3 | P4 | P5 | P6 | P7 | P8 | P9 | P10 | P11 | P12 | P13 | P14 |
|---|---|---|---|---|---|---|---|---|---|---|---|---|---|---|---|
| Mutagenicity | nMAE | 0.071 ±0.018 | 0.046 ±0.008 | 0.074 ±0.009 | 0.048 ±0.009 | 0.077 ±0.012 | 0.066 ±0.013 | 0.057 ±0.010 | 0.072 ±0.011 | 0.043 ±0.010 | 0.071 ±0.020 | 0.007 ±0.002 | 0.161 ±0.044 | 0.167 ±0.024 | 0.102 ±0.016 |
|  | AUC | 0.887 ±0.039 | 0.936 ±0.018 | 0.870 ±0.022 | 0.928 ±0.022 | 0.912 ±0.014 | 0.910 ±0.028 | 0.943 ±0.011 | 0.898 ±0.010 | 0.910 ±0.031 | 0.935 ±0.022 | 0.889 ±0.099 | 0.949 ±0.015 | 0.926 ±0.028 | 0.796 ±0.029 |
| MCF-7 | nMAE | 0.008 ±0.001 | 0.010 ±0.000 | 0.019 ±0.001 | 0.012 ±0.001 | 0.017 ±0.001 | 0.017 ±0.002 | 0.014 ±0.003 | 0.017 ±0.002 | 0.013 ±0.002 | 0.011 ±0.003 | 0.015 ±0.001 | 0.031 ±0.005 | 0.011 ±0.001 | 0.019 ±0.001 |
|  | AUC | 0.920 ±0.022 | 0.943 ±0.044 | 0.941 ±0.026 | 0.937 ±0.010 | 0.959 ±0.016 | 0.913 ±0.018 | 0.919 ±0.028 | 0.897 ±0.032 | 0.859 ±0.030 | 0.910 ±0.030 | 0.918 ±0.045 | 0.955 ±0.027 | 0.925 ±0.042 | 0.904 ±0.042 |
| ZINC | nMAE | 0.012 ±0.001 | 0.018 ±0.003 | 0.029 ±0.004 | 0.012 ±0.001 | 0.020 ±0.003 | 0.019 ±0.002 | 0.018 ±0.003 | 0.013 ±0.001 | 0.009 ±0.002 | 0.011 ±0.001 | 0.005 ±0.001 | 0.025 ±0.007 | 0.009 ±0.002 | 0.014 ±0.002 |
|  | AUC | 0.904 ±0.055 | 0.966 ±0.030 | 0.957 ±0.014 | 0.967 ±0.027 | 0.940 ±0.061 | 0.965 ±0.028 | 0.974 ±0.007 | 0.906 ±0.022 | 0.963 ±0.008 | 0.980 ±0.010 | 0.991 ±0.014 | 1.000 ±0.000 | 0.995 ±0.004 | 0.972 ±0.026 |
| ogbg-molhiv | nMAE | 0.002 ±0.001 | 0.001 ±0.001 | 0.001 ±0.000 | 0.017 ±0.020 | 0.005 ±0.003 | 0.021 ±0.024 | 0.002 ±0.001 | 0.003 ±0.002 | 0.004 ±0.003 | 0.002 ±0.000 | 0.004 ±0.002 | 0.011 ±0.004 | 0.002 ±0.001 | 0.003 ±0.001 |
|  | AUC | 0.919 ±0.021 | 0.958 ±0.013 | 0.923 ±0.024 | 0.919 ±0.022 | 0.914 ±0.038 | 0.928 ±0.007 | 0.975 ±0.011 | 0.882 ±0.057 | 0.915 ±0.040 | 0.923 ±0.017 | 0.961 ±0.028 | 0.917 ±0.033 | 0.972 ±0.017 | 0.902 ±0.020 |
| ogbg-molpcba | nMAE | 0.000 ±0.000 | 0.000 ±0.000 | 0.000 ±0.000 | 0.000 ±0.000 | 0.000 ±0.000 | 0.000 ±0.000 | 0.000 ±0.000 | 0.000 ±0.000 | 0.000 ±0.000 | 0.000 ±0.000 | 0.000 ±0.000 | 0.000 ±0.000 | 0.000 ±0.000 | 0.000 ±0.000 |
|  | AUC | 0.946 ±0.046 | 0.961 ±0.031 | 0.945 ±0.025 | 0.971 ±0.015 | 0.983 ±0.018 | 0.958 ±0.019 | 0.972 ±0.040 | 0.902 ±0.053 | 0.944 ±0.047 | 0.946 ±0.029 | 0.998 ±0.003 | 0.952 ±0.042 | 0.962 ±0.046 | 0.921 ±0.064 |
| Peptides-func | nMAE | 0.001 ±0.000 | 0.016 ±0.006 | 0.029 ±0.010 | 0.017 ±0.015 | 0.008 ±0.003 | 0.001 ±0.001 | 0.001 ±0.000 | 0.001 ±0.001 | 0.001 ±0.000 | 0.001 ±0.001 | 0.003 ±0.000 | 0.002 ±0.001 | 0.000 ±0.000 | 0.001 ±0.001 |
|  | AUC | 0.949 ±0.004 | 0.987 ±0.007 | 0.977 ±0.014 | 0.930 ±0.020 | 0.970 ±0.011 | 0.941 ±0.068 | 0.897 ±0.093 | 0.952 ±0.060 | 0.936 ±0.061 | 0.980 ±0.021 | 0.929 ±0.035 | 0.940 ±0.058 | 0.882 ±0.081 | 0.964 ±0.018 |
| PCQM-Contact | nMAE | 0.000 ±0.000 | 0.003 ±0.000 | 0.006 ±0.001 | 0.000 ±0.000 | 0.001 ±0.000 | 0.000 ±0.000 | 0.000 ±0.000 | 0.000 ±0.000 | 0.000 ±0.000 | 0.000 ±0.000 | 0.000 ±0.000 | 0.003 ±0.001 | 0.000 ±0.000 | 0.001 ±0.000 |
|  | AUC | 0.998 ±0.002 | 0.963 ±0.005 | 0.948 ±0.009 | 0.968 ±0.017 | 0.982 ±0.013 | 0.946 ±0.008 | 0.997 ±0.002 | 0.947 ±0.040 | 0.987 ±0.002 | 0.994 ±0.001 | 0.990 ±0.006 | 0.986 ±0.027 | 0.999 ±0.002 | 1.000 ±0.000 |

Table 5: Statistical properties of real-world molecular datasets. We report the number of graphs, as well as the average number of nodes and edges. All datasets are endowed with node and edge labels.

| Name | # Graphs | Avg. # nodes | Avg. # edges |
|---|---|---|---|
| Mutagenicity (Kersting et al., 2016) | 4337 | 30.32 | 30.77 |
| MCF-7 (Kersting et al., 2016) | 27770 | 26.40 | 28.53 |
| ZINC (Gómez-Bombarelli et al., 2018) | 12000 | 23.16 | 24.92 |
| ogbg-molhiv (Hu et al., 2021; Wu et al., 2018) | 41127 | 25.51 | 27.47 |
| ogbg-molpcba (Hu et al., 2021; Wu et al., 2018) | 437929 | 25.97 | 28.11 |
| Peptides-func (Dwivedi et al., 2022; Singh et al., 2015) | 15535 | 150.94 | 153.65 |
| PCQM-Contact (Dwivedi et al., 2022) | 529434 | 30.14 | 30.54 |

metrics, we used Area Under the Curve (AUC) as well as normalized Mean Absolute Error (nMAE). For AUC, we employed the One-vs-One approach to evaluate the model's performance across multiple classes. The nMAE is defined as the mean average error normalized by the true values. More formally, for true and predicted values $y_i$ and $\hat{y}_i$ with $i \in [N]$, it is given by $\frac{1}{N} \sum_{i=1}^{N} \frac{|y_i - \hat{y}_i|}{\max(1, |y_i|)}$ where the max function is used to avoid divisions by zero.

Table 4 presents the predictive performance across a wide range of dataset and pattern graph combinations. The results clearly demonstrate that subgraph counting on real-world molecular graphs can be done quite accurately.

## D.2 Number of truncated universal covers

In Table 6, we provide the numbers of distinct $\mathrm{WL}^\ell$ node labels for $\ell \in [1, 6]$, or equivalently, the numbers of non-isomorphic truncated universal covers $\eta_{\ell,\mathcal{G}}$ over a given dataset $\mathcal{G}$. We observe that, the Peptides-func dataset has low values of $\eta_{\ell,\mathcal{G}}$, wich suggests the applicability of the results of Section 4.

These values, as well as the values reported in Table 2 and Table 3, are obtained by taking into account edge label information. Moreover, as discussed in the previous section, we retain only the atom type as node features.

Table 6: Number of node and edge labels of common molecular datasets (where we restrict node labels to atom types only), and number $\eta_{\ell,\mathcal{G}}$ of non-isomorphic truncated universal covers.

| Dataset | node labels | edge labels | $\eta_{\ell,\mathcal{G}}$ | | | | | |
|---|---|---|---|---|---|---|---|---|
| | | | $\ell = 1$ | $\ell = 2$ | $\ell = 3$ | $\ell = 4$ | $\ell = 5$ | $\ell = 6$ |
| Mutagenicity | 14 | 3 | 334 | 4997 | 21118 | 43750 | 63568 | 76920 |
| MCF-7 | 46 | 3 | 668 | 18163 | 112803 | 229231 | 328616 | 409294 |
| ZINC | 21 | 3 | 499 | 13006 | 70302 | 144592 | 198159 | 229065 |
| ogbg-molhiv | 55 | 7 | 1923 | 37286 | 169766 | 319986 | 446262 | 547678 |
| ogbg-molhiv | 44 | 11 | 1185 | 44190 | 314139 | 915519 | 1857547 | 3040681 |
| Peptides-func | 6 | 5 | 76 | 411 | 1308 | 4704 | 18248 | 68392 |
| PCQM-Contact | 15 | 4 | 1745 | 97914 | 913960 | 2812873 | 5249611 | 7492474 |

## D.3 Quite-colorfulness in molecular datasets

We now experimentally evaluate the assessment made in Section 5.2 on the quite-colorfulness of subgraph isomorphisms in cases where the pattern at hand is not quite-colourful itself. Figure 6 reports the proportion of quite-colorful subgraph isomorphisms from non-quite-colorful tree patterns. Specifically, for a given dataset, we iterate over all subgraph isomorphisms from the pattern to the disjoint union of dataset graphs, and check each subgraph isomorphism for whether it is quite colourful. We report the ratio $|Q|/|S|$ of quite-colorful subgraph isomorphisms $|Q|$ to the total number of subgraph isomorphisms $|S|$ for increasing numbers of WL iterations $l$.

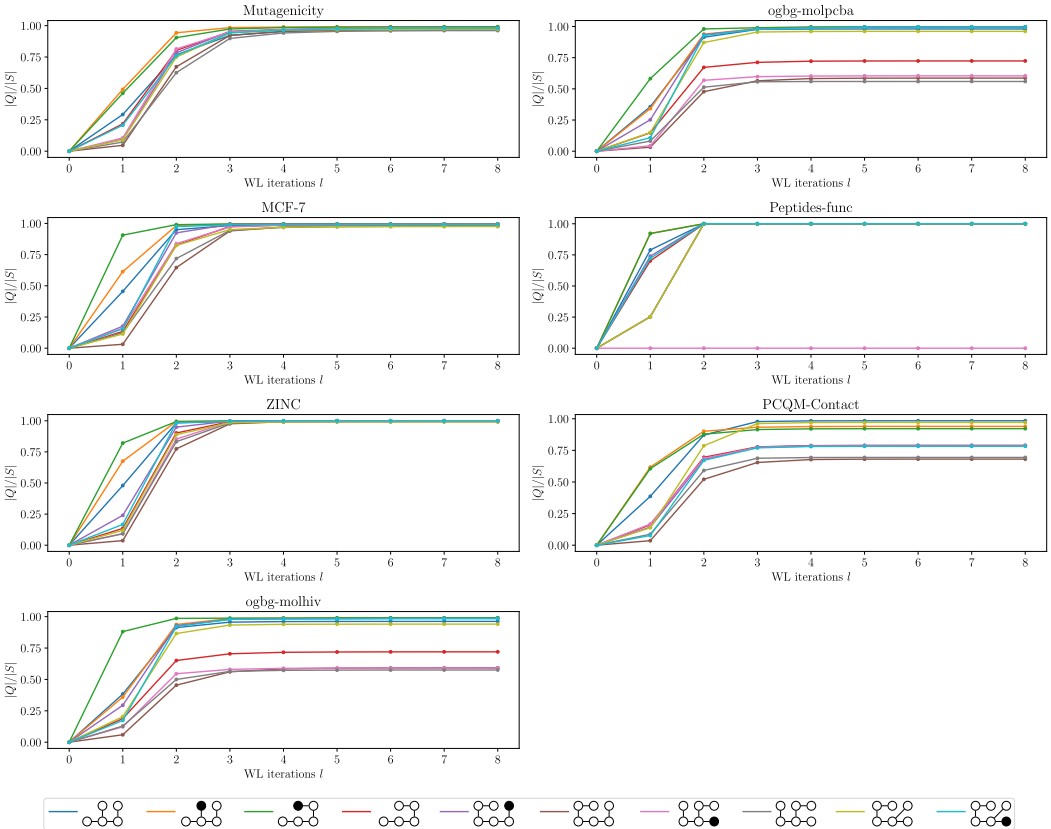

Figure 6: Proportion of subgraph isomorphisms that are quite colourful, reported for colors $c$ obtained by increasing numbers of color refinement iterations $l$. The proportion is given as $|Q|/|S|$, where $|Q|$ is the number of quite colourful subgraph isomorphisms from the pattern to the dataset graphs, and $|S|$ is the total number of subgraph isomorphisms.

The results show that for several real-world datasets, such as MCF-7, Mutagenicity, and ZINC, nearly all subgraph isomorphisms are quite colourful when target graphs are node-colored using 4 iterations of color refinement. Naturally, as the patterns themselves are not quite-colorful, none of the subgraph isomorphisms are quite-colorful when target graphs are not node-colored with color refinement (case $l = 0$). Notably, an interesting case arises in the Peptides-func dataset, where for one of the tree patterns none of the subgraph isomorphisms are quite colourful, regardless of the value of $k$.

## D.4 RULING OUT STAR PATTERNS

We now empirically validate Corollary 2 and Theorem 5 on challenging synthetic datasets. More specifically, we demonstrate that (non-induced) subgraph counting can be done in practice for scenarios fulfilling parent-colorfulness (see Def. 5) and quite-colorfulness (see Def. 3). Recall that while Chen et al. (2020) show that subgraph counting cannot be done by GNNs in general, (non-induced) counting of star-shaped patterns nonetheless remains possible. To rule out the possibility that a model is leveraging this information in the following experiments, we specifically construct datasets where every graph has the *same multiset* of star-shaped patterns.

In all experiments, we used a 4-layer GNN, as specified by the functions in Section A.1, with a hidden dimension of 512, a batch size of 128, and the Adam optimizer. The data was split into 80% for training and 20% for testing. We report the predictive performance after 1,000 epochs.

To verify the claim in Corollary 2, we randomly generated 2,000 graphs, each with 32 nodes and 39 edges, ensuring that the minimum cycle length was 5. Node labels were assigned based on node degree to guarantee that the mappings from the patterns are parent-colorful. More precisely, we first

generated trees of size 32 using a fixed node degree sequence, then added 8 edges while ensuring that the resulting graphs have the same node degree set and contained no cycles of length 4 or less.

Table 7 (left) shows the predictive performance for several pattern graphs. In accordance with Corollary 2, the results show a near perfect predictive performance on the test set on all pattern graphs.

We furthermore investigate the case where pattern mappings are quite-colorful. For this, we generated a more challenging dataset of 2,000 randomly generated graphs with 96 nodes and 120 edges each, with no constraints on the minimum cycle length. More precisely, we randomly generated the graphs using a fixed node degree sequence. Node labels were assigned based on node degree. We consider patterns that ensure mappings are quite-colorful. Table 7 (right) shows that the model learns to count very accurately. Note that the predictive performance increases for larger patterns, which might be due to the usually large and diverse subgraph counts of small patterns.

Table 7: Predictive performance for non-induced subgraph counting in scenarios where pattern matching is parent-colorful (left) and quite-colorful (right).

| Metric | | | | | | Metric | | | | | |
|--------|-------|-------|-------|-------|-------|--------|-------|-------|-------|-------|-------|
| MAE | 0.000 | 0.020 | 0.000 | 0.090 | 0.015 | MAE | 0.128 | 0.138 | 0.015 | 0.000 | 0.000 |
| AUC | 1.000 | 0.980 | 1.000 | 0.959 | 0.968 | AUC | 0.966 | 0.860 | 0.986 | 1.000 | 1.000 |

