# OpenReview forum: "Graph Neural Networks Can (Often) Count Substructures"
_ICLR.cc/2025/Conference — ICLR 2025 Spotlight_

### Official Review · Reviewer_gZ4p · 2024-10-29

**Soundness:** 3
**Presentation:** 4
**Contribution:** 3
**Rating:** 6
**Confidence:** 2

**Summary:**

This paper presents a solid theoretical framework on GNNs ability in subgraph counting, bridging the gap between their theoretical limitations and real-world performance. One of the key contributions is the identification of specific conditions under which GNNs can efficiently compute functions on graphs based solely on local substructures around the nodes. Additionally, the authors introduce a novel dynamic programming algorithm for problems related to subgraph isomorphism, showing that GNNs can effectively simulate this approach. They also back up their claims with experimental results, which adds to the paper's credibility.

Overall, I found the paper to be quite interesting. However, my current rating is a 5. I would be willing to raise my score if the authors address the weaknesses I’ve pointed out.

**Strengths:**

The paper is well-written and effectively motivates the problem it tackles. The authors provide a solid background that helps readers understand the context and importance of their work. Additionally, the authors use various synthetic and empirical datasets to strengthen their analysis and showcase the practical relevance.

**Weaknesses:**

Here are some weaknesses I noticed in the paper that I would like the authors to address:

- The authors did not reference important works in the field, such as "Counting Graph Substructures with Graph Neural Networks" by Kanatsoulis and Ribeiro, and "Improving Graph Neural Network Expressivity via Subgraph Isomorphism Counting" by Bouritsas et al. A more thorough literature review is needed.

- Additionally, the paper lacks comparisons with other methods and does not discuss how existing methods could be adapted for their task.

- Finally, the absence of code limits the reproducibility of their results. I appreciate the provided pseudocode, as it makes the paper easier to follow.

**Questions:**

I refer the authors to the weaknesses.

---

> ### Author Response · Authors · 2024-11-21
> **Response**
>
> Thank you for recognizing the clarity and the importance of our work.
>
> We believe that, by addressing the weaknesses that you pointed out, we made our submission more solid, and we therefore thank you for that.
> Please let us know if you have any further or follow-up questions so that we can try to address them.
>
> > The authors did not reference important works in the field, such as "Counting Graph Substructures with Graph Neural Networks" by Kanatsoulis and Ribeiro, and "Improving Graph Neural Network Expressivity via Subgraph Isomorphism Counting" by Bouritsas et al. A more thorough literature review is needed.
>
> Thank you for pointing out these references. We have included a paragraph to discuss these, and other references given by the other reviewers, in Section 1.1 and in Section 5.2.
> In particular, we discuss the relationship with Kanatsoulis and Ribeiro. The paper also proposes a method for GNNs to count substructures, but it's based on the assumption of the graphs having random features, which does not apply in general to standard GNNs.
>
> > Additionally, the paper lacks comparisons with other methods and does not discuss how existing methods could be adapted for their task.
>
> Please note that our work is the first one that discusses sufficient conditions that go beyond the worst-case analysis for standard message-passing GNNs to be able to count subgraphs. Because of this, our comparison with similar works can only be quite limited. Thanks to your suggestions though, we have added an in-depth comparison with the works that are the closest to ours.
> Indeed, we discuss in Section 1.1 and in Section 3 that the main difference of our work with the negative results of the seminal paper of Chen et al. is that we go beyond the worst-case analysis, which considers arbitrary target graphs. We have also added at the end of Section 5.1 a paragraph on how our results extend the results of Chen et al. (2020, Theorem 3.5) and of Zhang et al. (2024, Theorem 4.5).
>
>
>
>
> > Finally, the absence of code limits the reproducibility of their results. I appreciate the provided pseudocode, as it makes the paper easier to follow.
>
> The code and datasets are now available as supplementary material, and will be made public on GitHub.

---

> > ### Comment · Reviewer_gZ4p · 2024-11-25
> > **Response to the authors comments**
> >
> > Thank you for the comment. I will improve my score based on your feedback and that of other reviewers.

---

> > > ### Author Response · Authors · 2024-11-27
> > > **Response**
> > >
> > > Thank you for your valuable feedback. We really appreciate it.

---

### Official Review · Reviewer_oojv · 2024-10-30

**Soundness:** 3
**Presentation:** 3
**Contribution:** 2
**Rating:** 6
**Confidence:** 4

**Summary:**

This work studies whether and under which conditions graph neural networks (GNNs) can sample-efficiently count substructures. This is motivated by the observation that GNNs can count graph patterns with good accuracy on real-world datasets, despite the known poor subgraph counting ability at worst case. By leveraging the fact that counting patterns is local and node-decomposable, the authors prove that GNNs can perform subgraph counting with number of parameters independent of graph size, under the condition that nodes' ego-nets can be distinguished by WL test.  Furthermore, GNNs are shown to be algorithmically-aligned with a tree counting algorithm that finds locally injective homomorphism. Experiments demonstrate the effectiveness of subgraph counting on real-world molecular graphs.

**Strengths:**

1. A theoratical justification of subgraph counting power beyond worst case is provided.
2. Sample efficiency: the number of parameters is shown to be bounded, and an alignment to a subtree counting algorithm is also demonstrated. These provides clues for possible good generalization of GNNs for subgraph counting.
3. The paper is well-structured and easy to follow.

**Weaknesses:**

1. In Table 1, only results of four graph patterns are shown. It makes the evaluation stronger if more results of common substructures can be provided, or an explanation of why the current chosen ones are important to consider. For example, 5-cycle is also a common substructure in molecules.
2. The algorithm alignment is largely restricted to tree patterns, which is less interesting especially for molecular graphs.

**Questions:**

1. Though Table 3 shows the high identifiable ratio of Mutagenicity dataset, I wonder why the counting performance (MAE) is poor  in Table 1.
2. To let the subgraph counting algorithm 1 work, it requires the graph to have some inherent node colors to satisfy parent-colorful. Does that mean the algorithm (and thus GNNs) cannot perform such counting if the graph has no node attributes?

---

> ### Author Response · Authors · 2024-11-21
> **Response**
>
> Thank you for your insightful feedback.
> We hope that our answers and our paper revisions, which include additional experiments as you requested, adequately address all your comments. Please let us know if you have any further or follow-up questions so that we can try to address them.
>
> > In Table 1, only results of four graph patterns are shown. It makes the evaluation stronger if more results of common substructures can be provided, or an explanation of why the current chosen ones are important to consider. For example, 5-cycle is also a common substructure in molecules.
>
> We now provide more comprehensive results in Table 4 in Section D.1. The selected patterns are the most frequent ones across all datasets (using a frequency threshold of 25%) to ensure that subgraph counts are non-zero in a substantial number of graphs. Notably, the 5-cycle pattern is not among this set. Nonetheless we added it among the patterns we consider, for its importance in molecular graphs.
> Note that the central message, i.e., that GNNs can count substructure quite satisfactorily in practice, remains the same.
>
>
> > The algorithm alignment is largely restricted to tree patterns, which is less interesting especially for molecular graphs.
>
> We agree that the ultimate goal is to understand the mechanisms behind how GNNs count more complex patterns.
> Note however that our results on $(\ell,k)$-identifiability, substantiated by the results of Section 6.1, already give sufficient conditions for counting small cyclic patterns. We added a remark to make this explicit.
> Moreover, as we mention in Sections 5.3 and B.2, the DP algorithm and the corresponding GNN alignment can be extended, albeit not perfectly, to cyclic patterns.
> We believe that pursuing this extension, for which we lay the foundations here, is an interesting research direction, but it would be outside of the scope of this paper.
>
> > Though Table 3 shows the high identifiable ratio of Mutagenicity dataset, I wonder why the counting performance (MAE) is poor in Table 1.
>
> Note that the counting performance is not perfect, but it can be hardly described as poor. The GNN achieves on average >$0.9$ AUROC on the counting classification problem, and ~$0.5$ MAE, which means that the count is, on average, less than 1 off from the true count. Indeed, if we normalize the absolute error by the true count, we obtain (for Mutagenicity and the patterns of Table 1) the value ${\rm avg. [ (|estimatedCount - trueCount|)/(true Count)} ] \simeq 0.09$, that is the prediction is less than 10% off from the true value on average.
>
> The worse performance with respect to the other datasets can be explained by the fact that this dataset is the smallest one we consider, which could account for the worse generalization performance.
>
> > To let the subgraph counting algorithm 1 work, it requires the graph to have some inherent node colors to satisfy parent-colorful. Does that mean the algorithm (and thus GNNs) cannot perform such counting if the graph has no node attributes?
>
> On the contrary, the algorithm can work also with **no** node attributes. The coloring of the nodes is achieved via $l$ iterations of color refinement (or, in the case of GNNs, $l$ message passing layers). This is described at the beginning of Section 5.2, and we added a sentence to make explicit that no initial node features are needed.
>
> Note that for two nodes that have the same attribute, if they have neighborhood of different sizes, they would receive different colors after one iteration of color refinement, and this coloring would propagate with more and more color refinement iterations. In fact, Babai et al. showed that random graphs (even without node attributes) will have all nodes with distinct colors after a constant number of iterations with high probability.
> Therefore, in general, given sufficient irregularity in the graphs, the algorithm can correctly work even if the graphs are initially unattributed.

---

> > ### Comment · Reviewer_oojv · 2024-11-26
> > **Official Comment by Reviewer oojv**
> >
> > I would like to thank the authors for their responses and their efforts of providing new experiments, which addressed my concerns. I will maintain my original score.

---

> > > ### Author Response · Authors · 2024-11-27
> > > **Response**
> > >
> > > Thank you for your reply and for appreciating our efforts in improving the paper.

---

### Official Review · Reviewer_m1EJ · 2024-10-31

**Soundness:** 3
**Presentation:** 3
**Contribution:** 3
**Rating:** 6
**Confidence:** 4

**Summary:**

This paper explores the capability of GNNs to detect and count substructures in real-world datasets, despite known theoretical limitations in their expressive power. The authors challenge established assumptions by demonstrating that GNNs can perform subgraph counting tasks with surprising accuracy, even when conventional GNNs struggle to distinguish certain non-isomorphic graphs. The main contributions include deriving conditions under which GNNs can effectively count subgraphs by leveraging local structures, designing dynamic programming algorithms for solving subgraph isomorphism in restricted graph classes that GNNs can simulate, and empirically validating these theoretical findings across various datasets. By introducing the concept of $(l, k)$-identifiability, the paper bridges the gap between theory and practice, providing a framework that explains the unexpectedly robust subgraph-counting abilities of GNNs and highlighting their practical utility beyond theoretical constraints.

**Strengths:**

1. The paper introduces novel concepts regarding the identifiability of graphs through message passing, which builds on the existing theoretical framework of GNNs. The exploration of $(l, k)$-identifiability offers a fresh perspective on subgraph counting tasks.

2. The writing is generally clear and structured, with a logical progression from theory to application. Key concepts are well-defined, making the paper accessible to a broad audience within the machine learning and graph theory communities.

3. The findings contribute to the ongoing discourse around the capabilities and limitations of GNNs in solving graph-related problems. By demonstrating that GNNs can efficiently solve subgraph counting under specific conditions, the work is likely to influence future research directions in both theoretical and applied settings.

**Weaknesses:**

1. While the method is shown to work for subgraph counting, there is minimal discussion of its computational complexity, particularly for larger graphs or deeper GNN layers. Insight into time complexity would make the approach more practical.

2. This work mentions WL limitations but does not analyze upper bounds or scenarios where the proposed method may struggle with expressivity. A comparative analysis with existing expressive GNN models would provide a clearer understanding of when this approach is suitable.

3. Adding visual representations of key concepts, such as the identification of $(l, k)$-identifiable nodes could enhance reader understanding and engagement.

4. Missing references:

[1] C. I. Kanatsoulis and A. Ribeiro, Counting Graph Substructures with Graph Neural Networks, ICLR 2024.

[2] Huang et al. Boosting the Cycle Counting Power of GNNs with I²-GNNs, ICLR 2023.

[3] Raffaele et al. Weisfeiler and Leman Go Loopy: A New Hierarchy for Graph Representational Learning, ICLR 2024 Workshop.

[4] Wang et al. *N*-WL: A New Hierarchy of Expressivity for Graph Neural Networks, ICLR 2023.

Their inclusion could offer a more comprehensive view of how this method stands against similar research in GNN perspective.

**Questions:**

1. Can the authors provide more specific examples or scenarios where the condition of $(l, k)$-identifiability fails?  This could help clarify the limitations of their proposed framework.

2. How do the authors justify the reliance on WL classes in determining graph indistinguishability? Are there potential edge cases where this might not hold true, particularly in practical applications?

3. The paper mentions that the GNN may require up to $2^{(n-1)}$ message passing layers in the worst case. What are the empirical implications of this for large graphs? Are there any optimizations or heuristics that could be discussed to mitigate this?

4. If feasible, providing access to the code developed would enable others to replicate your experiments and potentially extend your work.

---

> ### Author Response · Authors · 2024-11-21
> **Response**
>
> Thank you for recognizing the novelty, the clarity and the importance of our work.
> We believe that, by addressing the weaknesses that you pointed out, we made our submission more solid, and we therefore thank you for that. Please find the detailed comments below.
>
> We hope that our revisions adequately address all your comments and positively influences your final score. Please let us know if you have any further or follow-up questions so that we can try to address them.
>
> > While the method is shown to work for subgraph counting, there is minimal discussion of its computational complexity, particularly for larger graphs or deeper GNN layers. Insight into time complexity would make the approach more practical.
>
> We added a discussion on the complexity of the algorithm in the paper at the end of Section 5.1, and a more detailed one in Section A.5. Note that for the GNN that simulates this, the complexity would depend on the number of layers and size of the MLP, which we characterize in Theorem 5 via the quantity $\zeta$, which is then upper bounded in the subsequent paragraph.
>
>
> > Adding visual representations of key concepts, such as the identification of -identifiable nodes could enhance reader understanding and engagement.
>
> Thank you for this suggestion, we indeed think visual representations could aid the reader, and we put significant effort in designing simple yet informative examples to put in our figures.
> We now provide example figures for (i) k-l-identifiable graph sets, (ii) quite-colorful maps and (iii) the execution of the dynamic program.
>
> > Missing references:
>
> Thank you for pointing out these references. We have included a paragraph to discuss these, and other references given by the other reviewers, in Section 1.1 and in Section 5.2.
>
> >This work mentions WL limitations but does not analyze upper bounds or scenarios where the proposed method may struggle with expressivity. A comparative analysis with existing expressive GNN models would provide a clearer understanding of when this approach is suitable.
>
> >Can the authors provide more specific examples or scenarios where the condition of identifiability fails? This could help clarify the limitations of their proposed framework.
>
>
>
> The first example of Figure 2 shows a case where the graph set is not (2,1)-identifiable. Interestingly, it is (3,1) identifiable. An example of a case where the condition fails for any $\ell$, is if the graphs in the dataset are all regular graphs. Then, the universal covers rooted at each node are all isomorphic, and the condition on $(\ell,k)$-identifiability would fail.
> However, as we show in Section 6.1, the conditions to make $(\ell,k)$-identifiability fail are rare in practice.
> We added a remark on this matter after Definition 2.
>
> Moreover, we added a remark at the and of Section 5.2 to clarify that on adversarial examples, such as regular graphs, also the dynamic programming algorithms, and thus the GNNs, will fail.
>
> > How do the authors justify the reliance on WL classes in determining graph indistinguishability? Are there potential edge cases where this might not hold true, particularly in practical applications?
>
> As shown in the classical results of Morris et al. (2019), message passing GNNs cannot distinguish non-isomorphic graphs that belong to the same WL class. Moreover, GNNs can, as shown in Xu et al.(2018) and Morris et al. (2019), distinguish graphs that belong to different WL classes. In practical applications though, GNNs might struggle with distinguishing all pairs of non-isomorphic graphs, but this does not seem to impact too much the task of subgraph counting, as shown in Table 1.
>
> > The paper mentions that the GNN may require up to $2^{n-1}$ message passing layers in the worst case. What are the empirical implications of this for large graphs? Are there any optimizations or heuristics that could be discussed to mitigate this?
>
> Note that we mention $2(n-1)$ layers, not $2^{n-1}$. The implications, as we mention in the paper, are that such GNNs would not be able to scale to large graphs.
> Note thought that we address this problem in the following section with the concept of $(\ell,k)$-identifiability. Indeed, for sets of graphs that satisfy this condition (and we show in Section 6.1 that this holds in practice), only $\ell$ layers are needed. Note that $\ell$ does not depend on the graph size, allowing the models to scale to large graphs.
> We have added a remark before Section 4.1 to clarify that we develop the concept of $(\ell,k)$-identifiability to use GNNs with number of layers and number of parameters independent of $n$.
>
> > If feasible, providing access to the code developed would enable others to replicate your experiments and potentially extend your work.
>
> Absolutely. The code and datasets are now available as supplementary material, and will be made public on GitHub.

---

> > ### Comment · Reviewer_m1EJ · 2024-11-22
> >
> > Thank you for your detailed and thoughtful responses to my questions. I appreciate the effort you have put into revising the paper and addressing the concerns I raised. I believe the revisions you have made have strengthened the submission. As such, I will maintain my original score.

---

> > > ### Author Response · Authors · 2024-11-27
> > > **Response**
> > >
> > > Thank you for your reply and for helping us improve the quality of the paper.

---

### Official Review · Reviewer_oWTR · 2024-11-03

**Soundness:** 3
**Presentation:** 4
**Contribution:** 3
**Rating:** 6
**Confidence:** 4

**Summary:**

In this paper, the authors argue that graph neural networks (GNNs) can often count graph patterns across various real-world datasets. They provide conditions under which GNNs can count subgraphs, based on local substructures around nodes. The authors also propose novel dynamic programming algorithms to solve the subgraph isomorphism problem for specific classes of pattern and target graphs and demonstrate that GNNs can efficiently simulate these algorithms.

**Strengths:**

**Originality**

The paper introduces several original ideas, including conditions under which GNNs can perform subgraph counting on specific class of graphs, dynamic programming algorithms for specific subgraph isomorphism problems, and an analysis of their connections with GNNs.

**Quality**

I have some concerns/questions regarding the quality of the paper (see "weaknesses").

**Clarity**

The paper is not easy to follow, primarily due to issues in presentation. For instance, some of the notations (e.g., Tree-LINc(u,h)[r]) are overly complex, lacking intuitive explanation, and certain concepts (e.g., pattern node, pseudo-dimension, etc.) are not clearly defined. The motivations for studying specific variants of the subgraph counting problem are not well-articulated, making it difficult to understand the significance of these variants within the broader context of the research.


**Significance**

The research area of graph neural networks and subgraph counting explored in this work is significant. However, the paper primarily focuses on WL-distinguishable graphs and tree patterns for its dynamic programming algorithms.  The contributions have a limited impact on the field.

**Weaknesses:**

(W1) The paper lacks technical depth and rigor.

Proposition 1 is limited to graphs that are distinguishable by the 1-WL test. Since 1-WL can distinguish only certain types of graphs, the results do not apply broadly to general subgraph counting functions. Rather than interpreting that GNNs can count all subgraphs within a class of 1-WL-distinguishable graphs, it would be more natural to explore whether GNNs can count specific classes of subgraphs, such as trees and forests (i.e., graphs with treewidth 1), within general graphs. If the class of graphs is restricted to those distinguishable by 1-WL, the results appear trivial.

The statement, *"It is immediate to see that subgraph counting and induced subgraph counting (i.e., counting the number of (induced) subgraph isomorphisms $\phi$ from $P$ to $G$) are node-decomposable,"* is central to the paper. However, if this is intended as a general statement, I have doubts about its correctness. I was unable to find a proof in either the main paper or the appendix. Since local neighborhood counts can lead to overcounting or miss global structures, they may not yield exact counts for general subgraph counting. Can the authors clarify this statement by formalizing it in the context of subgraph counting and providing either a formal proof or relevant citations? Please explain why this statement holds or does not hold for general subgraph counting.

The proof of Theorem 1 relies solely on node-level functions. However, if graph-level functions were considered, these two graphs could be easily distinguished, as they differ in the number of nodes and edges.

(W2) The significance of this work is unclear and appears to be limited.

Regarding the statement, "We develop novel algorithms solving the (non-induced) graph isomorphism problem on restricted pattern or target graph classes," can you clarify what is meant by the (non-induced) graph isomorphism problem? Generally, graph isomorphism is a different problem from subgraph isomorphism. Moreover, for restricted pattern or target graph classes, such as trees and forests, it is known that the expressivity of MPNN (and 1-WL) is upper bounded by homomorphism counts of forest [2]. Given this, why is there a need to develop dynamic programming algorithms to tackle the locally injective homomorphism problem specifically for tree patterns and to use a GNN model to simulate this? It would be helpful if the authors can explain how their approach differs from or improves upon existing methods for counting homomorphisms in trees and forests, and clarify the specific advantages or novel insights gained from their dynamic programming algorithms and GNN simulations compared to known results about MPNN expressivity.


(W3) Missing related work

The paper [2] specifically discuss how the expressivity of GNNs can be measured in a hierarchy of subgraph counting.


References:

[1] Beyond Weisfeiler-Lehman: A Quantitative Framework for GNN Expressiveness, Zhang, et al., ICLR 2024

[2] N-WL: A New Hierarchy of Expressivity for Graph Neural Networks, Wang, et al., ICLR 2023

**Questions:**

See the questions in "Weaknesses".

Below are some additional questions:

(1) The abstract mentions "sample-efficiently learn to count subgraphs." It is unclear what "sample-efficiently learn" means in this context. Can the authors provide a precise definition of "sample-efficiently learn" in this context, including any relevant metrics or benchmarks they use to measure sample efficiency?

(2) In Table 1, which GNN model is used? Appendix A.5 does not specify the GNN architecture. Also, what are the classes, and how is AUROC  calculated for induced subgraph counting in the multi-class classification setting? A statistical description of the datasets (e.g., number of graphs and classes) along with relevant citations would also be helpful.

(3) Regarding "the classical negative result of Chen et al." mentioned in the first paragraph of Section 3, can the authors specify which theorem in their paper this refers to?  In my review of the paper, it appears that Chen et al. just use counterexamples (a pair of graphs that have different induced-subgraph-counts of the pattern but cannot be distinguished from each other by 2-WL) to prove their results. I don't understand the claim "This result, however, requires that the set of graphs at hand features specific WL-indistinguishable graphs, which is unrealistic in practice". Can the authors clarify this claim?

(4) Assigning random colors to nodes in a graph would result in different colors for nodes within the same orbit. Detecting nodes in the same orbit is non-trivial and requires the computation of graph automorphisms. How would the work of color coding address this issue?

---

> ### Author Response · Authors · 2024-11-21
> **Response**
>
> We thank you for your insights on how the paper could be improved, and for acknowledging the high relevance of the field of work.
> Thanks to your feedback, we improved the formalism and incorporated a more thorough comparison with previous work and several experimental results to support our claims, which we believe have strengthened our submission.
>
> Please find below the answers to your comments. We hope that our answers and revisions address your concerns, and we’d be happy to address any additional questions you may have.
>
> > The paper is not easy to follow...
>
> Thank you for pointing out how to improve the clarity of our paper. We have modified the manuscript as follows:
> - now provide example figures for (i) k-l-identifiable graph sets, (ii) quite- and parent-colorful maps and (iii) the execution of the dynamic program.
> - we better describe the notation for the algorithm (lines 354 to 374), clarifying the inputs and the outputs.
> - we replaced "pattern nodes" with "pattern graph's nodes"
> - the pseudo dimension is formally defined in section A.2.
> -  We now moved the algorithm for the locally injective homomorphism problem to the appendix and focus on the (quite-colorful) subgraph isomorphism problem, which has a clearer motivation. The condition on quite-colorfulness is needed to go beyond the worst-case analysis, which would allow to count only star-shaped patterns.
>
> > However, the paper primarily focuses on WL-distinguishable graphs and tree patterns for its dynamic programming algorithms.
>
> Indeed our paper focuses in the first part on WL-distinguishable or $(\ell, k)$-identifiable graphs, but we also show in Section 6.1 that these conditions hold on several datasets. Our goal is indeed to find sufficient conditions for counting subgraphs, beyond the worst-case analysis.
> For the second part of the paper, the dynamic programs indeed work for tree patterns. While this result is limited, it is the first extension beyond the worst-case analysis of Chen et al. (2020, Theorem 3.5) and of Zhang et al., (2024, Theorem 4.5), see also reply to W2.
> Moreover, we discuss briefly in Section 5.3 and B.2 that the algorithms can be extended to work, albeit not perfectly, to cyclic patterns. This however is a major expansion, for which we lay the foundations here, and we are planning to explore this in future work.
>
> > Proposition 1 is limited to graphs that are distinguishable by the 1-WL test.
>
> We are a bit confused by this criticism.
> We indeed state this limitation in Proposition 1. Our goal is indeed to find sufficient conditions that hold in practice for which GNNs can count substructures, and WL distinguishability is one of them. As shown in Section 6.1, most real world graphs are WL distinguishable.
>
> Moreover, Proposition 1 is only the first simple result, what motivates the subsequent contributions. First, $(\ell,k)$-identifiability is a property that can hold even without 1-WL-distinguishability (e.g. for graphs where all k-ego-nets are isomorphic), and we show in Section 6.1 that it also holds in practice.
> Secondly, Section 5 discusses exactly what you mention in you comment, that is whether GNNs can count trees within general graphs. Our DP restricts the classes of pattern graphs, and can work also on WL-indistinguishable graphs.
>
> > The statement, "It is immediate to see...
>
> We have added a formal proof in the Appendix, and refer to it in the main paper.
> Note also that counting rooted subgraphs on each node of the target graph does not overcount the total number of subgraph isomorphisms.
> Note that we are counting the number of maps between the two graphs, so we are not dividing by the number of automorphisms of the pattern graph. In this formulation, it is easier to see the node decomposability. However, the result holds also in the formulation where one divides the number of maps by $|Aut(P)|$, by a simple rescaling.
>
> > The proof of Theorem 1 relies solely on node-level functions.
>
> Your observation is indeed correct, a GNN would be able to distinguish the two graphs, but we show that it won't be able to distinguish some pairs of nodes.
> The goal of Section 4 is to understand the scenarios where GNNs can count subgraphs at the node level. As we state in the paper, we do this to obtain GNNs that have a number of parameters independent on the graph size. Indeed, the GNNs that consider graph-level functions in Proposition 1 must have size that depends exponentially on the graph size. We have added a remark to highlight our goals: "*This gives hope for a GNN model that relies only on local structures...*".
> Moreover, counting (rooted) subgraphs at the node level is an interesting task itself, and our results help understand its hardness.
>
> > Generally, graph isomorphism is a different problem from subgraph isomorphism.
>
> Thank you for pointing this typo out. We meant "solving the (non-induced) **subgraph** isomorphism problem". We fixed this.
>
> .
>
> **We continue answering the questions in the comment below.**

---

> > ### Author Response · Authors · 2024-11-21
> > **Response (continued)**
> >
> > > Moreover, for restricted pattern or target graph classes, such as trees and forests...
> >
> > > (W3) Missing related work
> >
> > Thank you for pointing out the important reference (Zhang et al., 2024).
> > This work deals with arbitrary target graphs (i.e., the worst-case scenario). In fact, this work puts nicely into context our results of Section 5 (while it is orthogonal to the results of Section 4).
> > Zhang et al. proves that a GNN can count subgraph isomorphisms iff the spasm of the pattern is composed only by trees. Indeed, for patterns for with quite-colorfulness is guaranteed, this is the case. Therefore our results provide the first upper bounds on the parameter count and on the sample complexity of a GNN that can realize Theorem 4.5 of Zhang et al.
> >
> > However, our results of Section 5 go beyond this worst-case analysis. Indeed, even for patterns whose spasm includes cyclic graphs, which according to Theorem 4.5 of Zhang et al. cannot be counted, the condition on quite-colorfulness of the subgraph isomorphism maps can hold. Therefore, such subgraph isomorphism maps can be counted by GNNs.
> > Thanks to your suggestion, we checked whether the conditions for quite-colorfulness hold in commonly used real world datasets, and found that this is indeed the case. We believe that this result significantly strengthens the theoretical statements of the section, and we thank you for that.
> > We have added a paragraph at the end of Section 5.2 to include a comparison with previous work, and we included the experimental results on quite-colorfulness in Section 6.
> >
> > .
> >
> > **Questions:**
> >
> >
> > > (1) The abstract mentions "sample-efficiently learn to count subgraphs."
> >
> > Thanks for pointing out this lack of clarity. We expanded the paragraph before Theorem 3 to better explain that "sample efficiently" means that fewer training samples are required to learn the task. We now also better describe what we mean with sample complexity and its relationship to the pseudo dimension. Moreover, we provide formal definitions in Section A.2.
> >
> >
> > > (2) In Table 1, which GNN model is used? Appendix A.5 does not specify the GNN architecture. Also, what are the classes...
> >
> > Thanks for pointing this lack of clarity. We have added the following several detailed clarifications and implementation details, which are described in Section D. In the following, we briefly address your points:
> >
> > - We use the GNN specified in Section A.1, which is implemented by, e.g., PyG's GraphConv. We have added remarks on this both in Section 1 and in Section D.
> > - The learning problem was stated a classification task, where the classes correspond to discrete count values ranging from zero to the maximum count observed in the training set.
> > - For AUC, we employed the One-vs-One approach to evaluate the model's performance across multiple classes.
> > - We have now added an additional table giving an overview over the dataset properties (Table 5).
> >
> >
> >
> >
> > > (3) Regarding "the classical negative result of Chen et al."...
> >
> > Thank you for pointing out that the clarity on this matter can be improved.
> > We are referring to Theorem 3.3 in Chen et al. (2020), we now say this in the paper.
> > As you correctly point out, they use as counterexamples pair of graphs that have different induced-subgraph-counts of the pattern but cannot be distinguished from each other by 2-WL. However, the construction of these pairs is very specific, and such pairs of graphs are not found in real world datasets. First of all, one of the two graphs of the counterexamle pairs is always disconnected, which is very rare in practice. Secondly, we show in Table 2 that most graphs are actually WL distinguishable, and this is the case where our Proposition 1 applies.
> >
> > > (4) Assigning random colors to nodes in a graph would result ...
> >
> > You are indeed correct, using color refinement (and thus message passing) we cannot assign different colors to nodes in the same orbit (in fact, nodes in the same WL color class). The quite-colorfulness condition though allows for some nodes to have the same color if they are e.g. neighbors. We added a remark at the end of Section 5.2 to clarify that there exist adversarial examples where quite-colorfulness fails.
> > Finally, there is a line of work on breaking symmetries in graphs by assignign random features to nodes (Sato et al, 2021), (Murphy et al. 2019). We think that these approaches could be exploited in our setting and that it would be an interesting question for future work.

---

> > > ### Comment · Reviewer_oWTR · 2024-11-26
> > > **Thanks for the Rebuttal**
> > >
> > > I would like to thank the authors for their detailed explanations, which have addressed my concerns. The quality and clarity of the revised version have been considerably improved. I will raise my score to 6.
> > >
> > > One additional comment on Definition 1: for $\Sigma_{u\in V_G}g(G,u)$, should $V_G$ be $\mathcal{V}_G$, instead?

---

> > > > ### Author Response · Authors · 2024-11-27
> > > > **Response**
> > > >
> > > > Thank you for your positive feedback. We sincerely appreciated your detailed comments, as your insights allowed us to significantly improve our paper.
> > > >
> > > > Circa Definition 1: no, $V_G$ is correct. For each graph $G$, one would sum the contributions of $g$ from all its nodes $u \in V_G$. The function $g$ is defined indeed on $\mathcal{V}_\mathcal{G}$, i.e. the set of possible node orbits. We will add a remark.

---

### Official Review · Reviewer_4XuU · 2024-11-04

**Soundness:** 4
**Presentation:** 4
**Contribution:** 4
**Rating:** 8
**Confidence:** 4

**Summary:**

In the submitted manuscript, the authors study the ability of Graph Neural Networks (GNNs) to count substructures. So far a negative result, showing that not all substructures can always be counted by GNNs has been proven. The authors significantly extend our knowledge of this area by proving under what conditions all substructures in a graph can be counted by GNNs. They identify several drawbacks in this general result. They then make use of the fact that subgraph counting is an inherently local task and refine the conditions on the dataset to in turn improve upon their result. Furthermore, two dynamic programs are presented for subgraph counting in trees and graphs without short cycles, and it is shown that GNNs can simulate both these algorithms to successfully count subgraphs. Finally, the authors present empirical properties of real-world datasets, validating the applicability of their theoretical results, and study the predictive performance of a GNN on simulated data.

**Strengths:**

- The paper is very well-written and clear.
- The theoretical results are of clear importance to the field and present significant progress.

**Weaknesses:**

Please find further detail on my listed weaknesses in my questions below.

- A clear area for potential improvement of this work is the empirical analysis. Your results could be more broadly validated empirically. But I appreciate that not all papers need both a substantial theoretical and empirical contribution and the theoretical contribution here is very strong.

- In parts the writing is a bit dense and difficult to follow. Some examples may help to make the text more accessible.

- You should state more explicitly what GNN you used in your experiments.

**Questions:**

1] On several occasions in the empirical analysis you are imprecise on what GNN you use to investigate your hypotheses. I think the clarity of the paper would be improved if you stated more precisely what GNN you used. I am in particular referring the Line 59 and the discussion/caption of Table 1, as well as the experiments in Section 6.2, e.g., Line 479.

2] The paper you presented is one that the reader really has to work through. It is dense and complex. More patient explanation and more frequent examples would make for more pleasant reading. Examples would aid especially in the more advanced concepts in the preliminaries and in Section 5.

3] In your list of limitations of the result of Proposition 1, it would be appropriate to add that you require an impractically deep GNN, where the number of layers $\ell$ appears to be equal to the maximal number of nodes $n$ in any given graph in the dataset.

4] Your experiments could be improved and extended in a variety of ways.

4.1] In the main paper it is completely unclear how your synthetic dataset is constructed. The appendix also does not add much further detail on this. I believe you must be more precise on how you construct your synthetic dataset.

4.2] I do not understand why you do not provide empirical results for GNNs counting substructures on real-world datasets. It seems incomplete to me to provide details about the $(\ell,k)$-identifiability of real-world datasets and to then only validate your theory on simulated datasets.

4.3] Since the width, i.e., hidden dimension, of the GNN's readout function and the number of required message passing layers play a role in your theoretical results, it would be nice to complement your theory with an empirical ablation study on the impact of these parameters in practice.

4.4] It would be better if you referred to the appendices on the empirical aspects of your work, i.e., the ones accompanying Section 6 in the main paper. In general more details on the experimental setting, i.e., more discussion in Appendix A.6, and possibly also the required code to reproduce the experiments would be nice.

5] Minor comments:

5.1] Since your write-up is of such high quality, I want to suggest a few minor reformulations to aid in perfecting it. Please feel free to reject any of these suggested edits. Line 67: " conditions for which" -> "conditions under which"; Line 233 "such function are" -> "such functions are"; Line 469 "table 3" -> "Table 3"

5.2] In Line 121 I think you may have a typo in the definition of a rooted tree. Specifically, in the following: "$\text{children}(q) = \q$" I think it may be correct to consider "$\mathcal{N}(q)$" here.

5.3] The algorithm name "$\text{TREE-LIH}_c(u,h)[r]$" is never formally introduced in the text and the argument in angular brackets $r$ is only mentioned in Theorem 4. It may be nicer to introduce this algorithm name and to more consistently and explicitly define the arguments the algorithm takes.

5.4] The notation in the algorithms are not always defined, e.g., "return $\epsilon$", is never explicitly defined. It may be nicer to add a brief explanation of this. I furthermore noticed that Algorithm 3 is never explicitly mentioned in the text and it would be nicer to draw the reader's attention to it at some point by pointing it out.

5.5] The model that you define in Appendix A.1 appears to be similar to the GraphSage model [1] to me. Did I miss an important difference? If not then it may be nice to cite their work.

[1] Hamilton, W., Ying, Z. and Leskovec, J., 2017. Inductive representation learning on large graphs. Advances in neural information processing systems, 30.

---

> ### Author Response · Authors · 2024-11-21
> **Response**
>
> Thank you for recognizing the clarity and the potential impact of our work to the graph learning community.
>
> Moreover, we want to thank you for your very thorough and insightful review, which made us spot and address some sections of the manuscript that could be strengthened.
>
>
> Please find detailed answers to all of your questions below.
>
>
> > W1: A clear area for potential improvement of this work is the empirical analysis. Your results could be more broadly validated empirically. But I appreciate that not all papers need both a substantial theoretical and empirical contribution and the theoretical contribution here is very strong.
>
> We now provide more comprehensive results in Table 4 in Section D.1.
> Note that the central message, i.e., that GNNs can count substructure quite satisfactorily in practice, remains the same.
>
> > W2: In parts the writing is a bit dense and difficult to follow. Some examples may help to make the text more accessible.
>
> > 2] The paper you presented is one that the reader really has to work through. It is dense and complex. More patient explanation and more frequent examples would make for more pleasant reading. Examples would aid especially in the more advanced concepts in the preliminaries and in Section 5.
>
> Thank you for your suggestion. Wow provide example figures for (i) k-l-identifiable graph sets, (ii) quite- and parent-colorful maps and (iii) the execution of the dynamic program. We indeed think visual representations could aid the reader in understanding the hardest concepts of our paper, and we put significant effort in designing simple yet informative examples to put in our figures.
>
>
> > W3: You should state more explicitly what GNN you used in your experiments.
>
> > 1] On several occasions in the empirical analysis you are imprecise on what GNN you use to investigate your hypotheses. I think the clarity of the paper would be improved if you stated more precisely what GNN you used. I am in particular referring the Line 59 and the discussion/caption of Table 1, as well as the experiments in Section 6.2, e.g., Line 479.
>
> Thanks for pointing this lack of clarity. We use the GNN specified in Section A.1, which is implemented by, e.g., PyG's GraphConv. We have added remarks on this both in Section 1, in Section D (Additonal rexperimental results, which includes the old Section 6.2).
>
>
> > 3] In your list of limitations of the result of Proposition 1, it would be appropriate to add that you require an impractically deep GNN, where the number of layers
>  appears to be equal to the maximal number of nodes
>  in any given graph in the dataset.
>
> We agree. We now mention it in the paper.
>
> > 4] Your experiments could be improved and extended in a variety of ways.
>
> Thank you for suggesting these improvements. We have, indeed, significantly expanded our experimental section, with several new and extended experiments:
> - We now provide comprehensive empirical results for subgraph counting on real-world molecular datasets (see Section D.1), covering 8 datasets and 14 pattern graphs.
> - We now include an evaluation on the quite-colorfulness of subgraph isomorphisms for all 8 datasets, validating the assessment on quite-colorfullness in practice when the patterns themselves are not quite-colourful.
> - We revised the evaluation of parent- and quite-colorfulness on synthetic graphs and included a more detailed description of the synthetic dataset generation process.
> - The appendix now contains a table proving general information on the real-world datasets.
> - We now include the code as supplementary material.
>
> > 5] Minor comments:
>
> Thank you for your insights. We addressed 5.1, 5.2, 5.3 and 5.4. Note that the GNNs in A.1 are from Morrie et al. (2019), and not GraphSage. The difference is that we use a sum instead of a mean, that is needed for ensuring WL expressivty.

---

> > ### Comment · Reviewer_4XuU · 2024-11-24
> >
> > I want to thank the authors for their detailed response. All my concerns (except for Point 4.3] as far as I could tell) have been addressed. I still believe that this paper should be accepted and therefore choose to maintain my score.

---

> > > ### Author Response · Authors · 2024-11-27
> > > **Response**
> > >
> > > Thank you for valuing our work, and for giving us plenty of insights to improve our paper. We sincerely appreciated your review.
> > >
> > > Below, we provide some preliminary results on the influence of the number of GNN layers and the hidden dimension, as you suggest in point 4.3. The table shows the mean absolute error (MAE) for counting the first pattern reported in Table 1 (path of size 5) in the ZINC dataset. We observed similar trends across other cases we tested. The results indicate that the performance improves with an increasing number of layers, and that a single layer is clearly insufficient, as expected for a path of this size. Regarding the hidden dimension, the performance seems to saturate with larger values.
> > > In summary, the results may be explained by the fact that the GNN is only able to identify fragments of the pattern, and not the entire one, when a low hidden dimension and number of layers is used.
> > > We acknowledge that a more thorough empirical analysis would be needed to draw final conclusions. We will mention this as an interesting future research direction.
> > >
> > > | Layers | d=4  | d=8  | d=16  | d=32  | d=64  | d=128  | d=256 | d=512 |
> > > |:-----:|:-----:|:-----:|:-----:|:-----:|:-----:|:-----:|:-----:|:-----:|
> > > | 1 | 0.54 | 0.471 | 0.361 | 0.357 | 0.339 | 0.327 | 0.325 | 0.328 |
> > > | 2 | 0.372 | 0.239 | 0.191 | 0.124 | 0.095 | 0.098 | 0.105 | 0.098 |
> > > | 3 | 0.386 | 0.169 | 0.109 | 0.065 | 0.046 | 0.056 | 0.049 | 0.05 |
> > > | 4 | 0.191 | 0.143 | 0.059 | 0.059 | 0.044 | 0.038 | 0.039 | 0.062 |

---

### Author Response · Authors · 2024-11-21
**General response**

Dear reviewers,
we would like to express our appreciation for the several positive comments on our paper, recognizing (i) the novelty of our work, (ii) the clear importance of the results to the field, and (iii) that the paper is well-structured and easy to follow.

We would like to thank you for your insightful comments, which we believe have strengthened our paper considerably. In particular, based on your feedback, we included
- several visual representations that could aid the reader in understanding the hardest concepts of our paper
- paragraphs that discuss the relationship with previous works (Section 1.1 and Section 5.2),
- a more thorough empirical analysis, including a novel result that shows that quite-colorful maps are the vast majority in real world datasets.
We also include our code in the supplementary material, and we will make it public on GitHub after the review process.

Note that because, of the new content, we moved the algorithm on locally injective homomorphisms and the experimental evaluation on challenging synthetic datasets to the Appendix.

---

### Meta-Review · Area_Chair_X4D8 · 2024-12-20

**Metareview:**

This submission explores the capability of Graph Neural Networks (GNNs) to count substructures in graphs, despite known theoretical limitations in their expressive power. The authors challenge established assumptions by demonstrating that GNNs can perform subgraph counting tasks with surprising accuracy under certain conditions.

The main contributions of this paper are:
- **Deriving conditions for effective subgraph counting** : The authors identify specific conditions under which GNNs can efficiently count subgraphs, leveraging local structures around nodes. This includes the concept of (l,k)-identifiability (quite-colorfulness condition), which bridges the gap between theory and practice.
- **Designing dynamic programming algorithms**: The work defines a dynamic programming algorithm to solve the subgraph isomorphism problem for specific classes of pattern and target graphs, such as trees and graphs without short cycles. They demonstrate that GNNs can efficiently simulate these algorithms.
- **Empirical validation**: The work presents empirical properties of real-world datasets, validating the applicability of their theoretical results. Experiments demonstrate the effectiveness of subgraph counting on various datasets, including molecular graphs.

I believe this research contributes to our understanding of the capabilities and limitations of GNNs in graph analysis tasks. Overall, the discussion between the reviewer and the authors led to significant improvements in the paper.

The authors' willingness to address the identified weaknesses improved the relationship to prior work and the new experiments strengthen the claims. These include
- **Better examples**: Specific examples and scenarios where the condition of (l,k)-identifiability fails were provided, along with remarks on the practical implications for large graphs and optimizations to mitigate scalability issues.
- **Discussion of some experimental results**: In the rebuttal, authors explained the relatively poor counting performance (MAE) in Table 1 for the Mutagenicity dataset by pointing out that the dataset is the smallest, which could account for worse generalization performance.
- **Extra experiments**: In the rebuttal, the authors provided additional experiments (included in Table 4) that evaluate more graph patterns, including the 5-cycle, to strengthen the evaluation.
- **Improved theory:** In the rebuttal, the authors provided a formal proof for a for a statement regarding node-decomposability, clarifying its relevance to subgraph counting.

These and other changes should be added to the final version of the paper. I did not easily find any of the proposed changes in the revised PDF (next time, please mark changes with a different text color).

**Additional Comments On Reviewer Discussion:**

The discussion between authors and reviewers was great. The paper improved tremendously during rebuttal.

---

### Decision · Program_Chairs · 2025-01-22

Accept (Spotlight)